# Cross-regulation of viral kinases with cyclin A secures shutoff of host DNA synthesis

Boris Bogdanow[1,5,9], Max Schmidt[2,6,9], Henry Weisbach[2,7], Iris Gruska [2], Barbara Vetter[2], Koshi Imami [1,8], Eleonore Ostermann [3], Wolfram Brune [3], Matthias Selbach [1,4], Christian Hagemeier[2] & Lüder Wiebusch [2✉]

Herpesviruses encode conserved protein kinases (CHPKs) to stimulate phosphorylation-sensitive processes during infection. How CHPKs bind to cellular factors and how this impacts their regulatory functions is poorly understood. Here, we use quantitative proteomics to determine cellular interaction partners of human herpesvirus (HHV) CHPKs. We find that CHPKs can target key regulators of transcription and replication. The interaction with Cyclin A and associated factors is identified as a signature of β-herpesvirus kinases. Cyclin A is recruited via RXL motifs that overlap with nuclear localization signals (NLS) in the non-catalytic N termini. This architecture is conserved in HHV6, HHV7 and rodent cytomegaloviruses. Cyclin A binding competes with NLS function, enabling dynamic changes in CHPK localization and substrate phosphorylation. The cytomegalovirus kinase M97 sequesters Cyclin A in the cytosol, which is essential for viral inhibition of cellular replication. Our data highlight a fine-tuned and physiologically important interplay between a cellular cyclin and viral kinases.

[1] Research group "Proteome Dynamics", Max Delbrück Center for Molecular Medicine, 13125 Berlin, Germany. [2] Labor für Pädiatrische Molekularbiologie, Charité Universitätsmedizin Berlin, 13353 Berlin, Germany. [3] Heinrich Pette Institute, Leibniz Institute for Experimental Virology, 20251 Hamburg, Germany. [4] Charité-Universitätsmedizin Berlin, 10117 Berlin, Germany. [5] Present address: Research group "Structural Interactomics", Leibniz Forschungsinstitut für Molekulare Pharmakologie, 13125 Berlin, Germany. [6] Present address: Medizinische Klinik m.S. Hämatologie, Onkologie und Tumorimmunologie, Charité Universitätsmedizin Berlin, 12200 Berlin, Germany. [7] Present address: PenCef Pharma GmbH, 13509 Berlin, Germany. [8] Present address: Laboratory of Molecular & Cellular BioAnalysis, Kyoto University, 606-8501 Kyoto, Japan. [9] These authors contributed equally: Boris Bogdanow, Max Schmidt. ✉email: lueder.wiebusch@charite.de

Herpesviruses are a wide-spread family of large, double-stranded DNA viruses replicating within the nuclei of their host cells. Generally, herpesviruses are highly adapted to their hosts and persist in a latent mode of infection unless immunodeficiency provokes viral reactivation and disease[1]. They have diversified into three subfamilies: neurotropic α-herpesviruses, broadly infective β-herpesviruses, and lymphotropic γ-herpesviruses. Despite their widely different pathogenic properties and clinical manifestations, all herpesviruses share a common set of conserved core genes, mostly encoding essential structural components and replication factors[2]. Among the few core genes with regulatory function are the conserved herpesvirus protein kinases (CHPKs)[3], which are medically important both as drug targets[4] and as prodrug activating enzymes[5].

CHPKs belong to the group of viral serine/threonine kinases[6]. Although possessing a considerable sequence divergence, CHPKs have a number of common characteristics including autophosphorylation, nuclear localization, incorporation into the tegument layer of virus particles, and phosphorylation of other tegument proteins[6]. CHPKs target regulators of the DNA damage checkpoint[7,8], phosphorylate the translational elongation factor EF-1δ[9], and counteract the IRF3-dependent type I interferon response[10].

CHPKs were reported to preferentially target cellular cyclin-dependent kinase (CDK) phosphorylation sites[11,12]. In particular, the kinases of human β- and γ-herpesviruses show significant structural and functional homology to CDKs[13–16]. This has led to their designation as viral CDK-like kinases (v-CDKs)[13]. V-CDKs lack amino acids that are known to be essential for interaction of cellular CDKs with cyclins, CDK inhibitors (CKIs) and CDK activating kinases (CAKs). Accordingly, v-CDKs are considered immune to cellular control mechanisms[17], despite the fact that the UL97 kinase of human cytomegalovirus (human herpesvirus 5, HHV5) can physically interact with various cyclins[18]. V-CDKs mimic CDK1 and 2 in phosphorylating Lamin A/C[13,14], retinoblastoma-associated tumor suppressor RB1 (refs. [13,17,19]), and deoxynucleoside triphosphate (dNTP) hydrolase SAMHD1 (refs. [20,21]). Collectively, the CDK-like activities of CHPKs facilitate the nuclear egress of virus capsids[22] and secure the supply of dNTPs and cellular replication factors for viral DNA replication[20,23].

Significant knowledge has accumulated over the recent years about CHPK functions. It is known that CHPKs can recruit cellular proteins as substrates via conserved docking motifs[19,24]. Further, CHPKs can be regulated by their interaction partners and even exert kinase independent functions[25,26]. However, a systematic analysis of CHPK protein–protein interactions is lacking and data linking interactions to functions of the kinase are scarce. To broaden our understanding of CHPKs, we used an affinity purification–massspectrometry (AP–MS) approach and identified shared and differential interaction partners of CHPKs from all herpesvirus subfamilies. We found that Cyclin A–CDK complexes build a common set of interactors for β-herpesvirus v-CDKs. Taking mouse cytomegalovirus (MCMV, murid herpesvirus 1 (MuHV-1)) as a model system, we show that during infection the stoichiometric formation of cytoplasmic v-CDK and Cyclin A assemblies causes a global shift in substrate phosphorylation and the viral shutoff of host DNA synthesis. Collectively, our data demonstrates that herpesvirus kinases have evolved as protein-interaction hubs that can recruit a rich repertoire of cellular proteins. The functional versatility of β-herpesvirus v-CDKs is underpinned by their ability to cross-talk with a cellular cyclin via direct, non-catalytic interaction.

## Results
**CHPKs target key regulators of transcription and replication.**
Systems-level approaches have provided important insights into the molecular functions of CHPKs. For example, yeast 2-hybrid screens revealed several binary binding partners of CHPKs[27,28] and phosphoproteome profiling was successfully used to assess CHPK substrate specificity[7,12,29,30]. However, comprehensive and comparative information about CHPK interaction partners at the proteome level is lacking.

To identify CHPK interaction partners, we transfected SILAC (stable isotope labeling of amino acids in cell culture) heavy and light labeled HEK-293T cells with HA-tagged CHPKs of seven different human herpesviruses (HHVs) or vector controls. We performed these experiments in triplicates, including label-swaps (Fig. 1a), and subjected the samples to HA affinity purification (HA-AP) and shotgun proteomics. We specifically detected the baits in extracts of transfected cells and in eluates of HA-APs (Supplementary Fig. 1a, b). The baits were among the most abundant proteins in the eluates (Supplementary Fig. 1c), consistent with good enrichment. We observed an overall good reproducibility of SILAC fold-changes for the individual replicates (Supplementary Fig. 2a). To discriminate candidate interactors from background binders, they were required to fall below a p-value of 0.05 and meet a SILAC fold-change cut-off at 1% false discovery rate (FDR). (Fig. 1b, Supplementary Fig. 2b, c). We quantified about 1500–2000 proteins in individual samples (Supplementary Fig. 1d) and classified 135 proteins as candidate interaction partners to at least one kinase (Supplementary Data 1). This list includes several previously found CHPK interactors and substrates, such as SAMHD1 (ref. [20]), RBL2 (ref. [31]), PPP2CA, PUM1, PRDX1 (ref. [7]), and NUMA1, TUBA1B, MAP7D1 (ref. [30]).

We next aimed to determine interaction partners that were common to all kinases, class-specific (i.e., restricted to either kinases of α, β, or γ-classes) or unique to individual kinases. Therefore, we clustered the SILAC fold-changes of candidate interactors across all tested kinases (Fig. 1c). We found the chaperonin containing TCP1 and the kinase maturation complex (CDC37, HSP90) to co-purify with all kinases analyzed. This confirms the previous observation that CHPKs interact with the same set of cellular proteins that assist in folding and maturation of cellular kinases[32,33].

Proteins that co-purified with BGLF4 (HHV4, Epstein–Barr-virus), but were not significantly enriched with other kinases, include factors involved in DNA replication (HERC2, E4F1, PCNA, SSBP1) and chromatin silencing (SMCHD1, CABIN1, HIRA, BEND3). Proteins that co-purified with pUL97 (HHV5) but not other kinases were functionally related to regulation of transcription (PHC2, RING1, CBX4, TRIM28, ZNF136, ZNF791). To validate our approach, we performed an additional SILAC AP–MS experiment directly comparing proteins enriched to BGLF4 and pUL97 (Supplementary Fig. 3a). Again, we found selective enrichment of the specific sets of host interactors described above to either BGLF4 or pUL97 (Supplementary Fig. 3b, Supplementary Data 2). By performing co-immunoprecipitation (co-IPs) experiments, we were able to validate the transcriptional regulators SPOP, CCAR2, BMI1, and TRIM28 as specific interactors of pUL97 (Supplementary Fig. 3c–f).

Importantly, we found that all human β-herpesvirus kinases co-purify with S/G2 phase-specific cyclins (Cyclin A, gene symbol: CCNA2, Cyclin B, gene symbol: CCNB1), CDKs (CDK1, CDK2), and associated proteins (SKP1, SKP2). While enrichment of Cyclin A–CDK2 was reproducibly strong, Cyclin B–CDK1 did not enrich with pUL97 in our transfection experiment (Fig. 1d). We were able to validate the interaction with Cyclin A–CDK2 by reverse co-IPs. Also, we observed this type of interaction for the homologous M97 kinase of MuHV-1, better known as MCMV (Fig. 1e). Collectively our interactome of human CHPKs provides a rich resource and suggests

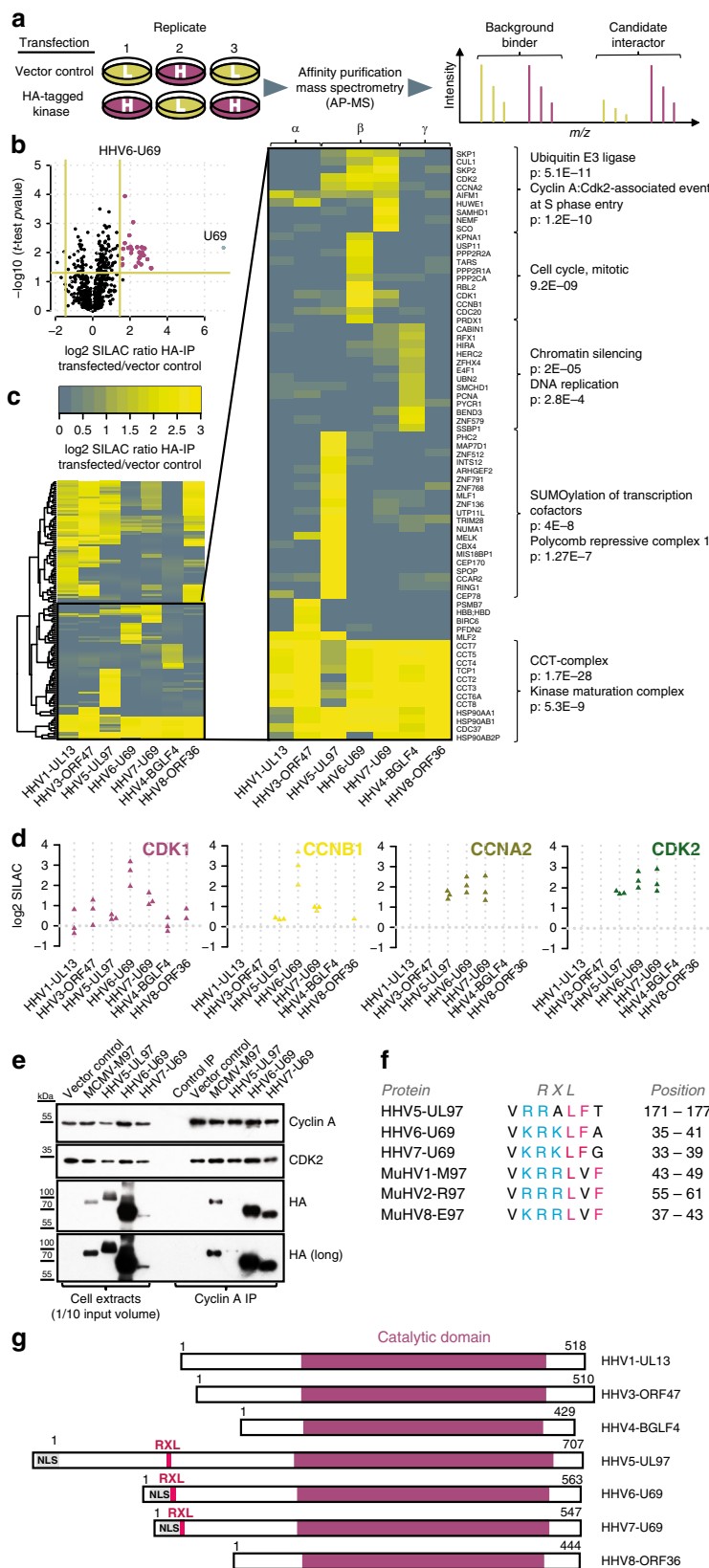

that these kinases are crucially involved in the regulation of transcription, epigenetic remodeling, and cell cycle control. Cyclin A–CDK2 complexes build a common subset of interactors for β-herpesvirus kinases, suggesting important functional implications.

**β-herpesvirus kinases bind cyclin–CDKs via NLS-RXL modules.** β-herpesvirus kinases lack most of the residues of CDKs that directly interact with cyclins, including a conserved PSTAIRE helix[17]. Instead, we found RXL/Cy motifs in the N-terminal, noncatalytic parts of β-herpesvirus kinases (Fig. 1f–g). Such motifs

**Fig. 1 CHPKs target key regulators of cellular replication and transcription. a** Experimental setup. SILAC heavy and light labeled HEK-293T cells were transfected with constructs encoding HA-tagged versions of human herpesviral kinases or control vectors in triplicates ($n = 3$). Samples were subjected to anti-HA affinity purification followed by mass-spectrometry. **b** Candidate selection. Candidate interaction partners were distinguished from background binders based on a combination of the $t$-test $p$-value of three replicates and the mean of the SILAC ratios. $P$-value cut-off at 0.05 and fold-change cut-off at FDR = 1%. A volcano plot for the HHV6-U69 kinase is exemplarily shown. **c** Cross-comparison of kinase interaction partners. All candidate interaction partners were compared for co-enrichment across the indicated kinases of α-, β-, and γ-HHVs. Enrichment profiles were clustered and selected sets of clusters were subjected to GO enrichment (inlet and brackets). **d** Cyclin–CDK complexes co-purify with β-herpesviral kinases. Depicted are the SILAC ratios of individual replicates ($n = 3$) for prey proteins CDK1, CCNB1, CCNA2, and CDK2 across AP–MS samples of the indicated kinases. **e** Validation of Cyclin A–CDK binding to β-herpesviral kinases. Whole cell extracts from transfected 293T cells were subjected to Cyclin A co-IP and analyzed by immunoblotting for the presence of Cyclin A, CDK2, and HA-tagged kinases. The immunoblots are representative of three independent experiments with similar results. **f** Alignment of putative Cyclin A binding sites in human and rodent β-herpesvirus kinases. Basic residues are highlighted in blue, bulky hydrophobic residues in pink. **g** Schematic of human herpesviral kinases. Shown are the location of the conserved catalytic domains and the relative positions of RXL/Cy motifs (pink) and predicted or experimentally validated nuclear localization signals (NLS) (gray) within the less-conserved N-terminal parts of the proteins.

are typically used for substrate and inhibitor recruitment to cyclin–CDKs[34]. Importantly, the positions of the RXL-type sequences within the largely divergent N termini are well conserved (Supplementary Fig. 4). These putative Cyclin A binding elements in β-herpesvirus kinases of the roseolovirus (HHV6, HHV7) and muromegalovirus (MuHV1, MuHV2, MuHV8) genera are in close proximity to clusters of positively charged residues (Supplementary Fig. 4), which are predicted, and in the case of HHV6 validated, classical bipartite nuclear localization signals (NLS)[35]. In fact, the C-terminal part of the NLS sequences directly overlaps with the N-terminal part of the RXL motifs (Fig. 2a). Thus, when we set out to test the contribution of RXL motifs in U69 and M97 to Cyclin A binding, we had to consider the possibility that RXL mutations may negatively affect NLS function. We therefore designed two mutant versions of each kinase: one disrupting the core of the RXL motif (RXL–>AXA) and one changing only the hydrophobic part (LF–>AA, LXF–>AXA), leaving the basic residues of the NLS intact (Fig. 2a). Both mutations abolished binding of M97 and U69 kinases to Cyclin A (Fig. 2b). Thus, RXL sequence motifs found in β-herpesvirus kinases act as Cyclin A docking sites. RXL mutation not only prevents Cyclin A binding but also interaction with other cyclin-associated factors found in the interactome analysis (Fig. 1c), as exemplarily shown for HHV6-U69 (Supplementary Fig. 5). This indicates that those factors are not direct v-CDK interactors but instead co-recruited with Cyclin A. Thus, the RXL motif triggers the formation of higher-order v-CDK–cyclin–CDK complexes.

We then assessed the consequences of RXL mutations for NLS function. To this end, we cloned 32–37 amino acid segments encompassing the overlapping NLS and RXL elements of M97 and U69 kinases into an NLS reporter construct[36] (Fig. 2c). Integration of wild-type (WT) NLS-RXL regions into the chimeric reporter induced nuclear accumulation of the otherwise cytoplasmic GFP signal (Fig. 2d, e), indicative of a functional NLS. RXL to AXA mutations weakened this activity in HHV6-U69 and even disrupted it in M97. By contrast, the NLS function remains intact when only the hydrophobic residues of the RXL/Cy motifs are mutated (Fig. 2d, e). These results demonstrate that cyclin binding and NLS sequences are integrated into a composite motif that can be functionally separated by mutations.

**M97 assembles cyclin–CDK complexes in infected cells.** We then aimed to analyze the NLS-RXL module in a representative infection system and chose MCMV for ease of manipulation. We first assessed the time-resolved protein interactome of M97 using SILAC and AP–MS (Supplementary Fig. 6a, Supplementary Data 3). Early during infection (12 h), the M97 interactome consisted almost exclusively of cyclins, CDKs, and associated

proteins (Supplementary Fig. 4b). In the late phase (36 h), additional viral and cellular factors co-purified with M97 (Supplementary Fig. 4d), including M50–M53, the nuclear egress complex of MCMV[37], and Lin54, the DNA-binding subunit of the cell cycle regulatory MuvB complex[38], which is known to be regulated by pUL97 (ref. [39]). Taken together, this indicates that M97 functions in cell cycle regulation, viral egress, and control of gene expression.

In consistency with the data from transfected cells, M97 interacted with Cyclin A/B–CDK complexes throughout infection (Fig. 3a, Supplementary Fig. 6b,d). In particular, Cyclin A–CDK2 was present in similar molar amounts as M97 itself in HA-M97 MS samples (Supplementary Fig. 6c, e), suggesting a strong and stoichiometric interaction in infected cells. Mutation of the RXL/Cy motif (Supplementary Fig. 5) disrupted the interaction of Cyclin A with M97 (Fig. 3c, d). However, these mutations did not influence Cyclin A protein levels (Fig. 3b) or Cyclin A-associated kinase activity (Fig. 3f). Moreover, RXL/Cy mutations, in contrast to the "kinase dead" K290Q mutation[23], did not compromise M97 levels and activity (Fig. 3b, e). Thus, M97–Cyclin A binding has no influence on abundance and enzymatic activity of the involved interaction partners. This is consistent with the view that β-herpesviral CHPKs are cyclin-independent kinases[17].

**Cyclin binding controls subcellular localization of M97.** In respect of the overlapping NLS and RXL/Cy motifs within M97, we next tested whether Cyclin A binding alters the subcellular localization of M97. To address this, we chose a system where Cyclin A levels are dynamically changing. In quiescent cells, Cyclin A is absent at very early time points of infection (Fig. 3b) and induced between 6 and 24 h by MCMV (Fig. 3b). Using immunofluorescence microscopy, we observed that M97-WT is nuclear at 8 h post infection (Fig. 4a), when Cyclin A is still low, but redistributes to a predominantly cytoplasmic localization pattern at 24 and 48 h post infection (Fig. 4a, middle left column), when Cyclin A levels are high. Thus, the appearance of M97 in the cytosol correlates with increased Cyclin A expression.

Then we tested the contribution of the RXL/NLS module to the time dependent change in M97 localization. When the NLS function was disrupted by R45A/L47A mutation, M97 was confined to the cytoplasm throughout infection (Fig. 4a, b, middle right column). By contrast, when the NLS part of the NLS-RXL module was left intact and only Cyclin binding was prevented (L47A/F49A), M97 showed a constitutively nuclear localization pattern (Fig. 4a, b, right column). Thus, Cyclin A binding is responsible for the late relocalization of M97. This provides evidence for a scenario, where Cyclin A masks the NLS and thereby effectively interferes with nuclear entry of M97.

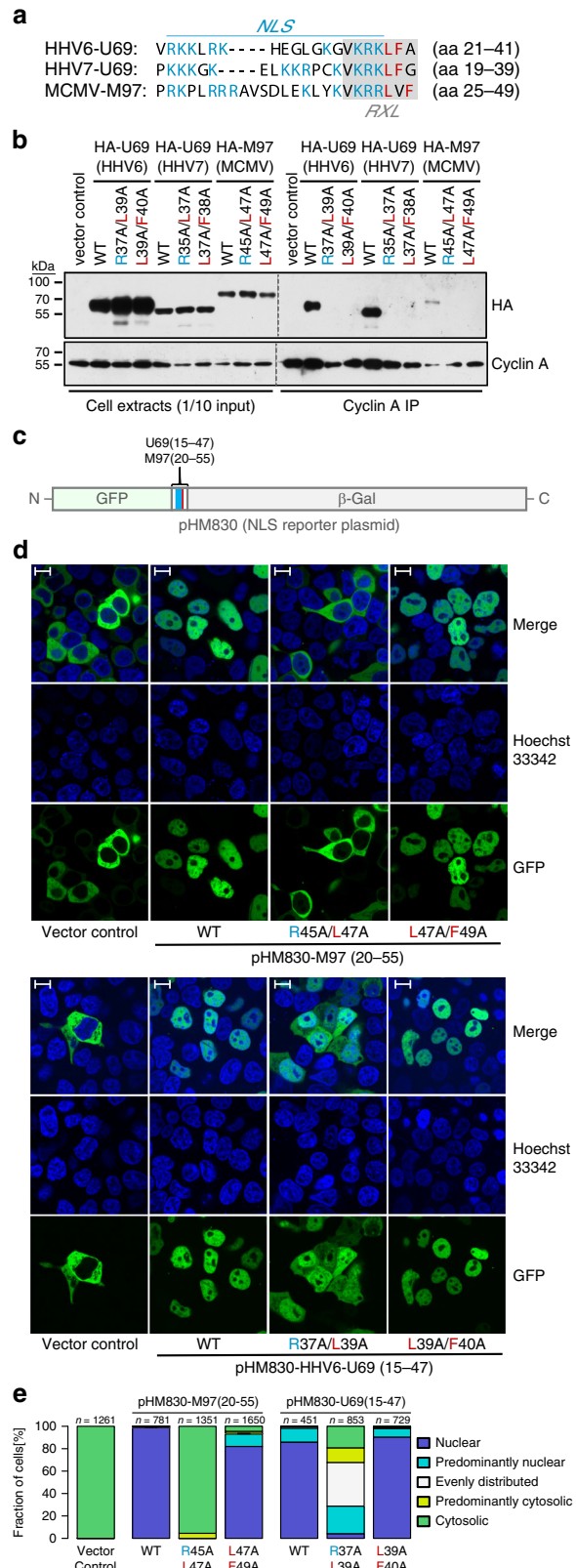

**Fig. 2 NLS and RXL/Cy motifs overlap in U69 and M97 kinases.**
**a** Sequence alignment of overlapping NLS-RXL elements in M97 and U69 kinases. Stretches of basic residues characteristic for nuclear localization signals (NLS) are highlighted in blue, conserved leucine and phenylalanine residues participating only in the RXL/Cy motif in pink. A mutation strategy was designed to separate NLS function from cyclin binding. RXL-AXA mutations affect both RXL and NLS sequences, LXF-AXA and LF-AA mutations only the RXL/Cy motif. **b** HEK-293T cells were transfected with HA-tagged wild-type (WT) or mutant forms of U69 and M97 kinases. Cyclin A co-IP was carried out at 2 days post transfection and analyzed by immunoblotting for the presence of viral kinases and Cyclin A. The immunoblots are representative of three independent experiments with similar results. **c** Sequence fragments encompassing the NLS-RXL region of U69/M97 mutant and wild-type kinases were cloned between and in-frame with the green fluorescent protein (GFP) and β-galactosidase genes in pHM830. **d**, **e** 293T cells were transfected with the M97/U69-NLS reporter constructs, as indicated. The subcellular localization of GFP was analyzed at 24 h post transfection using confocal live-cell imaging microscopy. Nuclei were counterstained with Hoechst-33342. Scale bars: 10 μm. **e** The indicated number of cells were categorized based on the subcellular localization of the GFP reporter relative to the Hoechst stain.

We globally assessed substrate phosphorylation by performing a proteomic analysis of phosphopeptides and whole cell lysates of SILAC-labeled cells at 24 h post infection (Supplementary Fig. 8a, Supplementary Data 4).

First, we corrected phosphosite ratios for protein level changes (Supplementary Fig. 8b, c). Next, we categorized proteins and corresponding phosphosites based on their GO annotation as nuclear, cytoplasmic or unclear ("no category") (Supplementary Fig. 8d). Then, we specifically analyzed phosphosites that match known v-CDK target motifs[12], such as pSP, pSXXK, LXpSP (p denotes the phosphorylated residue). When cells were infected with M97$^{R45A/L47A}$ mutant, pS/TP and pSXXK sites residing in the cytosol were significantly stronger phosphorylated than nuclear sites (Fig. 4c). By contrast, when cells were infected with the M97$^{L47A/F49A}$ mutant, the same set of sites were stronger phosphorylated when they belonged to nuclear proteins. The most pronounced differences in the target spectrum were observed when phospho-serines followed by prolines were positioned between hydrophobic amino acids and lysines (Supplementary Fig. 8e). Collectively, these data argue that Cyclin A binding enables switch-like changes in the substrate spectrum and subcellular distribution of M97.

**M97 inhibits host DNA synthesis by Cyclin A sequestration.** The binding of M97 to Cyclin A may have functional consequences not only for M97 but also for Cyclin A. Cyclin A is essential for cellular DNA replication and cell cycle progression from S phase to mitosis[40]. Importantly, Cyclin A function depends on its nuclear localization[41,42]. Therefore, we analyzed whether M97 binding impacts the subcellular distribution of Cyclin A. We found Cyclin A, and to a lesser extent CDK2, to be depleted from the nucleus in MCMV-WT-infected cells (Fig. 5a). This effect was Cyclin A specific as Cyclin E was evenly distributed between nuclear and cytoplasmic fractions. Mutation of the M97 translation start site prevented the cytoplasmic enrichment of Cyclin A. Mutations of the RXL/Cy motif in M97 even led to a predominant nuclear localization of Cyclin A. Thus, Cyclin A is sequestered within the cytosol dependent on its binding to M97.

This observation prompted us to investigate whether the nuclear depletion of Cyclin A by M97 affects cell cycle

Next, we studied how the dynamic and Cyclin A-dependent relocalization of M97 impacts protein phosphorylation. Therefore, we compared cells that were either infected with RXL deficient MCMV-M97$^{L47A/F49A}$ or NLS-RXL deficient MCMV-M97$^{R45A/L47A}$. The two mutant viruses reflect the early nuclear or late cytosolic localization of M97 during MCMV-WT infections.

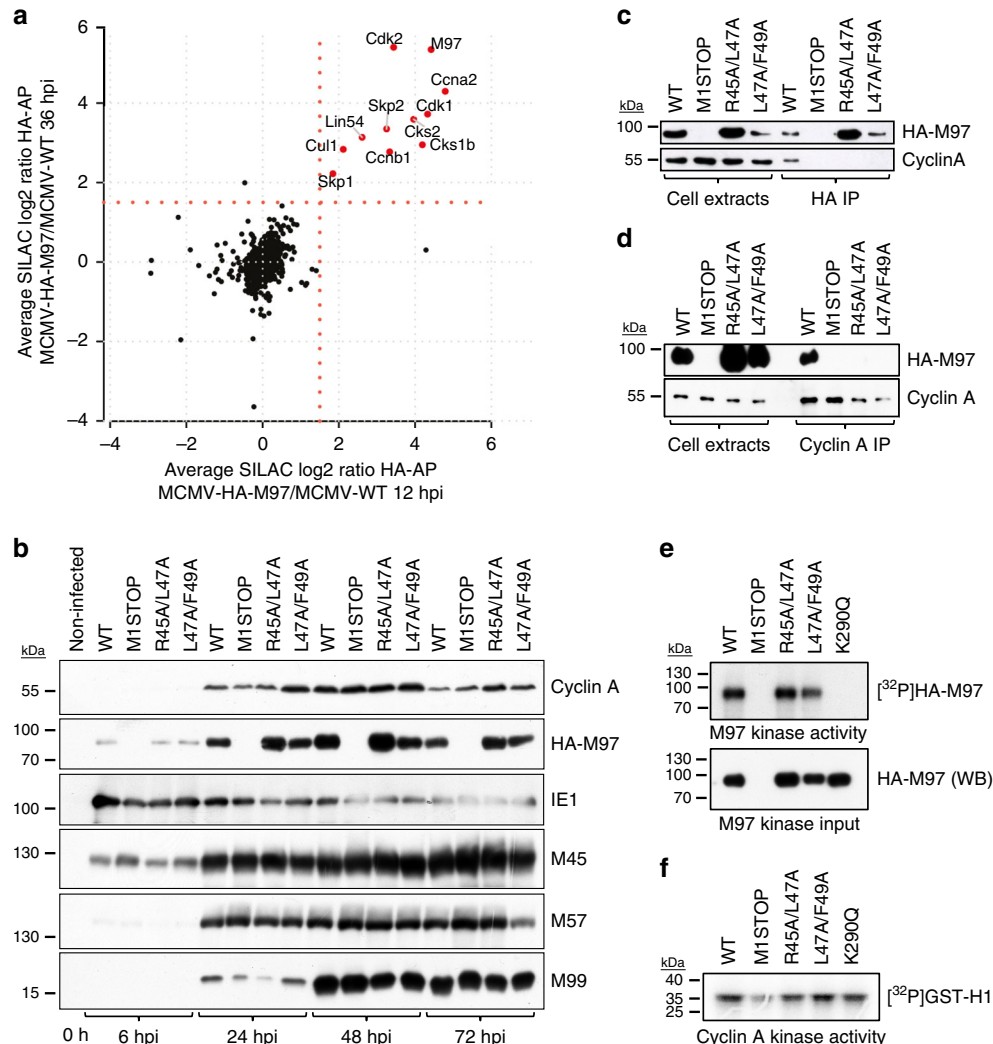

**Fig. 3 Formation of M97–cyclin–CDK assemblies during infection. a** A time-resolved interactome of M97 during MCMV infection (Supplementary Fig. 6). Enrichment ratios for proteins co-purifying with HA-M97 at 12 or 36 h post infection in a scatterplot. The cut-off at SILAC fold-change 1.5 is indicated by a red dotted line. The mean of $n = 3$ replicates is depicted. **b**–**f** Serum-starved 3T3 fibroblasts were infected with the indicated MCMV-HA-M97 variants. **b** The expression of Cyclin A and of selected immediate-early, early and late MCMV gene products was monitored by immunoblot analysis. Affinity purification of the indicated HA-M97 variants by anti-HA-coupled paramagnetic microbeads (**c**) or reverse co-IP (**d**) confirms RXL/Cy-dependent interaction with Cyclin A–CDK2. Assays for M97 (**e**) and Cyclin A (**f**) enzymatic activity. Lysates from MCMV-infected 3T3 fibroblasts were subjected to anti-HA-IP (**e**) or Cyclin A-IP (**f**) and used as input material for γ-P32-ATP kinase assay. Recombinant Histone H1 was used as RXL/Cy-independent Cyclin A substrate and autophosphorylation of M97 was measured for the M97 kinase assay. Incorporation of γ-P32-ATP was visualized by autoradiography. The immunoblots (**b**–**f**) and autoradiograph (**f**) are representative of three independent experiments with similar results.

progression. Therefore, we measured the DNA content of quiescent fibroblasts infected with WT and M97 mutant viruses by flow cytometry (Fig. 5b, c, Supplementary Fig. 9). We found a ganciclovir-sensitive increase of viral DNA in MCMV-WT-infected cells, consistent with previous reports[43]. By contrast, M97 mutant viruses caused a rapid and ganciclovir-resistant accumulation of cells with a G2/M DNA content. The latter phenotype was reversed by stable and efficient knockdown of Cyclin A (Supplementary Fig. 10). These observations demonstrate that loss of M97–Cyclin A interaction causes S phase entry and cell cycle progression to G2/M phase.

We aimed to confirm the crucial role of M97–Cyclin A interaction for inhibition of cellular DNA synthesis by an orthogonal approach. We chose to combine EdU pulse labeling and fluorescence microscopy to spatially discriminate sites of active viral or cellular DNA synthesis. In MCMV-WT-infected cells, DNA synthesis is confined to nuclear replication compartments that stain positive for the viral single-stranded DNA-binding protein M57 (Fig. 5d, e). By contrast, cells infected with M97-RXL/Cy mutants incorporated EdU in foci distributed over the whole nucleus, indicating that the restriction of cellular DNA replication was lost. Thus, M97–Cyclin A complex formation serves as a mechanism to shut off competing host DNA synthesis during productive infection.

Very similar results were obtained in primary, non-immortalized mouse embryonic fibroblasts (MEF), the only difference being that here the RXL/Cy mutation leads to a cellular DNA content that exceeds the normal 4n DNA content of diploid G2/M cells (Supplementary Fig. 11). This indicates that the M97 mutant virus bypasses cellular control mechanisms protecting primary cells from DNA over-replication.

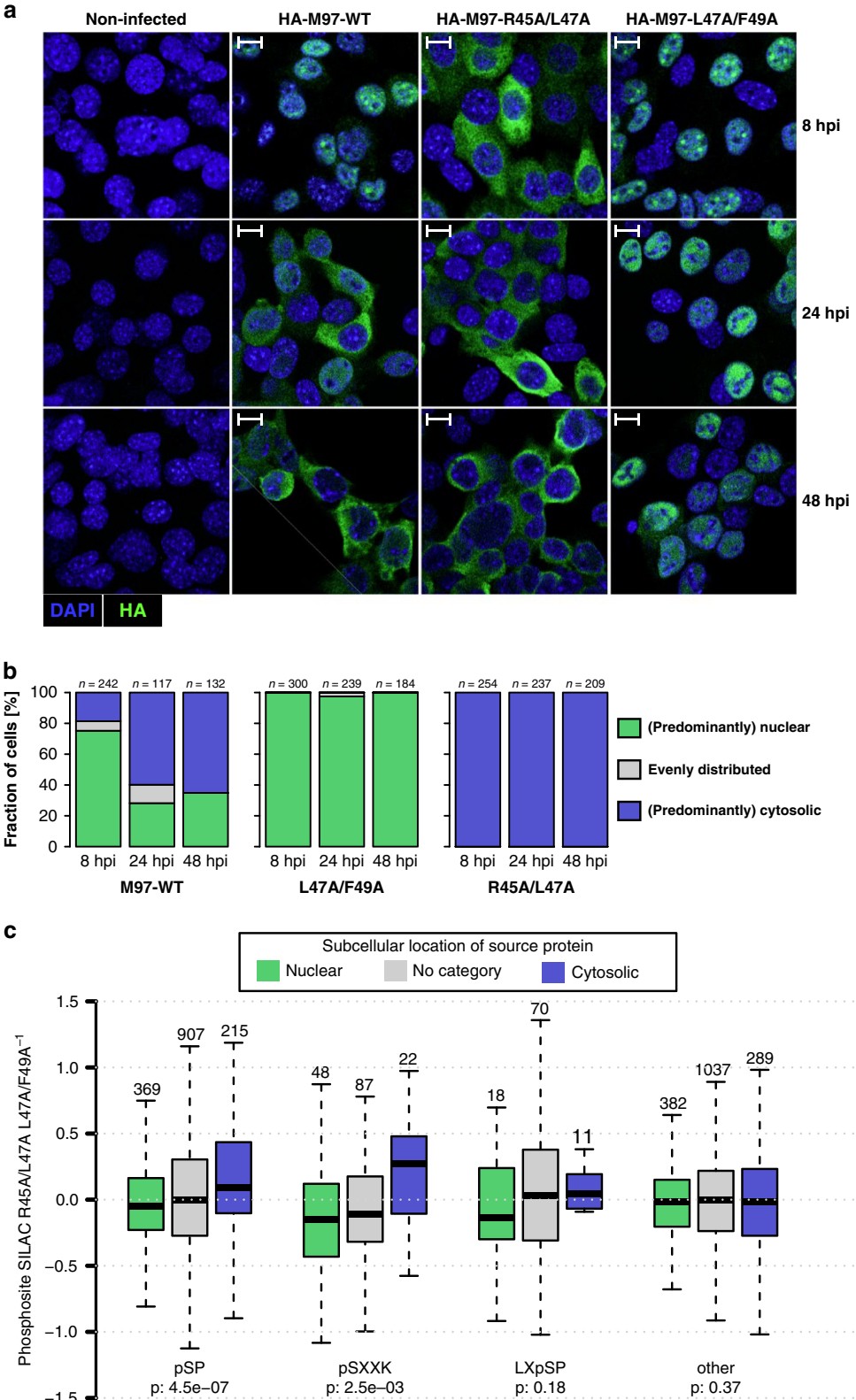

Although M97 is not essential for MCMV in vitro, its deletion negatively affects virus growth[44]. When we tested our set of M97 mutants on MEFs, a virus growth curve analysis revealed that the NLS-RXL module has a larger impact on virus replication than the kinase activity itself (Fig. 5f). This underlines the importance of nonenzymatic protein–protein interactions for the proper function of conserved β-herpesvirus kinases during the virus life cycle.

## Discussion

Here, we present the first comparative analysis of CHPK inter-actomes. Our analysis led to the identification of host–protein interaction signatures that are common, class-specific or unique to CHPKs (Fig. 1c). The interaction of β-herpesvirus-CHPKs with Cyclin A orchestrates higher-order complex formation of cell cycle regulatory factors (Fig. 3). Notably, the interaction is

**Fig. 4 The RXL-NLS module controls M97 localization and substrate phosphorylation. a** Serum-starved 3T3 fibroblasts were infected with the indicated MCMV-HA-M97 variants or left uninfected. At 8, 24, and 48 h post infection, cells were examined by confocal immunofluorescence microscopy for subcellular localization of HA-M97 (green). Nuclei were counterstained with DAPI. Scale bars: 10 µm. Representative images are shown. **b** The indicated number of cells were categorized based on the subcellular localization of HA-M97 relative to the DAPI stain. **c** SILAC-labeled 3T3 cells were serum starved and infected with NLS-RXL/Cy-deficient M97$^{R45A/L47A}$ or RXL/Cy-deficient M97$^{L47A/F49A}$ viruses (Supplementary Fig. 6). At 24 h post infection cells were harvested and subjected to a phosphoproteomic workflow. Boxplots for phosphosites that belong to proteins with nuclear, cytoplasmic, or uncategorized subcellular GO annotation are depicted with their SILAC log2 fold-change between M97$^{R45A/L47A}$ and M97$^{L47A/F49A}$ infection. pSP, pSXXK, LXpSP, or all other sites were assessed individually. Center line, median; box limits, upper and lower quartiles; whiskers, 1.5× interquartile range; outliers (data points outside the whiskers) were removed for visibility. *P*-values are based on an one-sided Wilcoxon rank-sum test comparing nuclear and cytosolic subsets. Data represent the mean of $n = 2$ replicates.

regulatory and has functional consequences for both, the CHPK and Cyclin A. For MCMV, it results in the dynamic relocalization of the viral kinase (Fig. 4a) and consequently an altered substrate spectrum (Fig. 4b). Further, the interaction leads to cytosolic sequestration of Cyclin A, which is essential for viral arrest of DNA replication (Fig. 5).

V-CDKs are a subset of CHPKs that share key aspects of host CDK-function and the ability to complement for CDK activity in yeast cells[13,14,17]. V-CDKs have lost sequence features enabling control by cellular factors, such as CAK, CKI, and cyclins[17]. Instead, we show that some v-CDKs acquired RXL motifs (Fig. 1), which are typically used by cellular CDK substrates and inhibitors for recognition by cyclins[34,45]. These motifs enable a regulatory cross-talk of Cyclin A and v-CDKs, which is characterized by cell cycle dependent regulation of the viral kinase on the one hand (Fig. 4) and neutralization of Cyclin A in stoichiometric protein assemblies on the other hand (Fig. 5, Supplementary Figs. 6 and 10, 11). The latter effect of v-CDK–Cyclin A interaction is reminiscent of CKI function. Thus, we propose that β-herpesviruses integrate the antipodal activities of CDKs and CKIs on one gene product. This combination allows a β-herpesvirus kinase like M97 to activate S/G2 metabolism[23] while inhibiting cellular DNA synthesis (Fig. 5).

An exception among human β-herpesviruses is HCMV, which has evolved a different gene product for neutralization of Cyclin A. HCMV produces the small protein pUL21a, which targets Cyclin A for proteasomal degradation[46,47]. Therefore, it seems that HCMV has shifted its Cyclin A-antagonistic, CKI function from its kinase to pUL21a. In that context it is interesting that the RXL motif in HCMV-pUL97 does not overlap with an NLS[48], suggesting an alternative function of pUL97–Cyclin A interaction (Fig. 1).

Short linear motifs (SLiMs) can be rapidly acquired by viruses and other pathogens to target host proteins[49,50]. Here, we found that within HHV6, HHV7, and rodent CMV kinases two such SLiMs are fused into one regulatory sequence element (Fig. 2). The physical overlap of RXL/Cy and NLS motifs is facilitated by their sequence composition as both contain contiguous stretches of basic amino acids. NLS motifs function as docking sites for nuclear import factors, mainly importin-α[51]. Accordingly, a docking competition mechanism makes binding to Cyclin A and nuclear localization of M97 mutually exclusive, enabling switch-like changes in viral kinase function (Fig. 6a). This puts β-herpesvirus kinases in a row with a number of cellular and viral proteins known to control nucleo-cytoplasmic localization via intermolecular NLS-masking, with NF-κB as the best understood example[52].

Remarkably, composite NLS-RXL/Cy elements are apparent in a number of key regulatory proteins of the host cell cycle. Although independently described, NLS and RXL/Cy motifs are overlapping in RB1 (refs. [53,54]), CDKN1A (also known as p21)[45,55], E2F1 (refs. [56,57]), and CDT1 (refs. [58,59]) (Fig. 6b). This feature is highly conserved in vertebrate orthologues (Supplementary Fig. 12). For

these cellular factors, it could be important to consider a docking competition between Cyclin A and importins, a possibility which is so far unexplored. Specifically, cyclin binding may be essential for the cell cycle dependent localization of these proteins[57,60,61]. In other cases, such as CDC6, RXL motifs are adjacent to but not overlapping with NLS. There, Cyclin A recruits CDKs to neutralize the NLS by phosphorylation[62].

In addition to the functionally important interaction of β-herpesvirus kinases with Cyclin/CDKs, our unbiased proteomic survey indicates that CHPKs interact with many more cellular proteins (Fig. 1c, Supplementary Data 1). Consistent with previous reports, many CHPK interaction partners are functionally related to DNA repair[7] and cell cycle control[13]. Remarkably, we found that CHPKs also interact with a variety of prominent transcriptional repressor complexes. For example, pUL97 interacts with TRIM28-ZNF complexes (Fig. 1c, Supplementary Fig. 3e), known to silence CMV gene expression in stem cells depending on the phosphorylation status of TRIM28 (ref. [63]). In addition, pUL97 co-purifies with PHC2, RING1, CBX4, and BMI1 (Fig. 1c, Supplementary Fig. 3f), all members of the poly-comb repressive complex 1, which was recently linked to control of HCMV replication[64]. Likewise, we found BGLF4 to interact with core subunits of the HIRA histone chaperone complex (HIRA, CABIN1, and UBN2). This complex restricts lytic infection by depositing histone H3.3 on incoming herpesvirus genomes[65]. Collectively, these data indicate that CHPKs can target host-derived master regulators of viral transcription, governing the decision between lytic and latent infection programs. This could allow tegument-delivered CHPKs to actively influence the outcome of herpesvirus infections.

Our interactome analysis indicates that CHPKs target master regulators involved in host replication, DNA repair, and transcription. It is important to note that this type of experiment cannot resolve protein interactions that depend on the environment of an infected cell. For example, CHPKs may be differentially abundant, differently modified, differently folded, or differently localized in transfected compared to infected cells. It is thus critical to interpret our findings in the broader context of kinase-associated functions that dynamically change during infection. For instance, phosphorylation of the nuclear protein SAMHD1 could be maintained by the M97 kinase early during infection[23] when M97 localizes nuclear (Fig. 4a). During later stages of infection, the nuclear lamina is locally disassembled and egress of viral capsids to the cytoplasm occurs. The shift from nuclear replication to cytosolic assembly is accompanied by the concurrent relocalization of the viral kinase. Thus, Cyclin A interaction helps the kinase to exert infection stage-specific functions.

## Methods

**Cells.** HEK-293T cells and NIH-3T3 fibroblasts were cultivated in Dulbecco's modified Eagle medium (DMEM) supplemented with 10% fetal (293T) or newborn (3T3) bovine serum, 2 mM L-alanyl-L-glutamine, 100 U mL$^{-1}$ penicillin, and

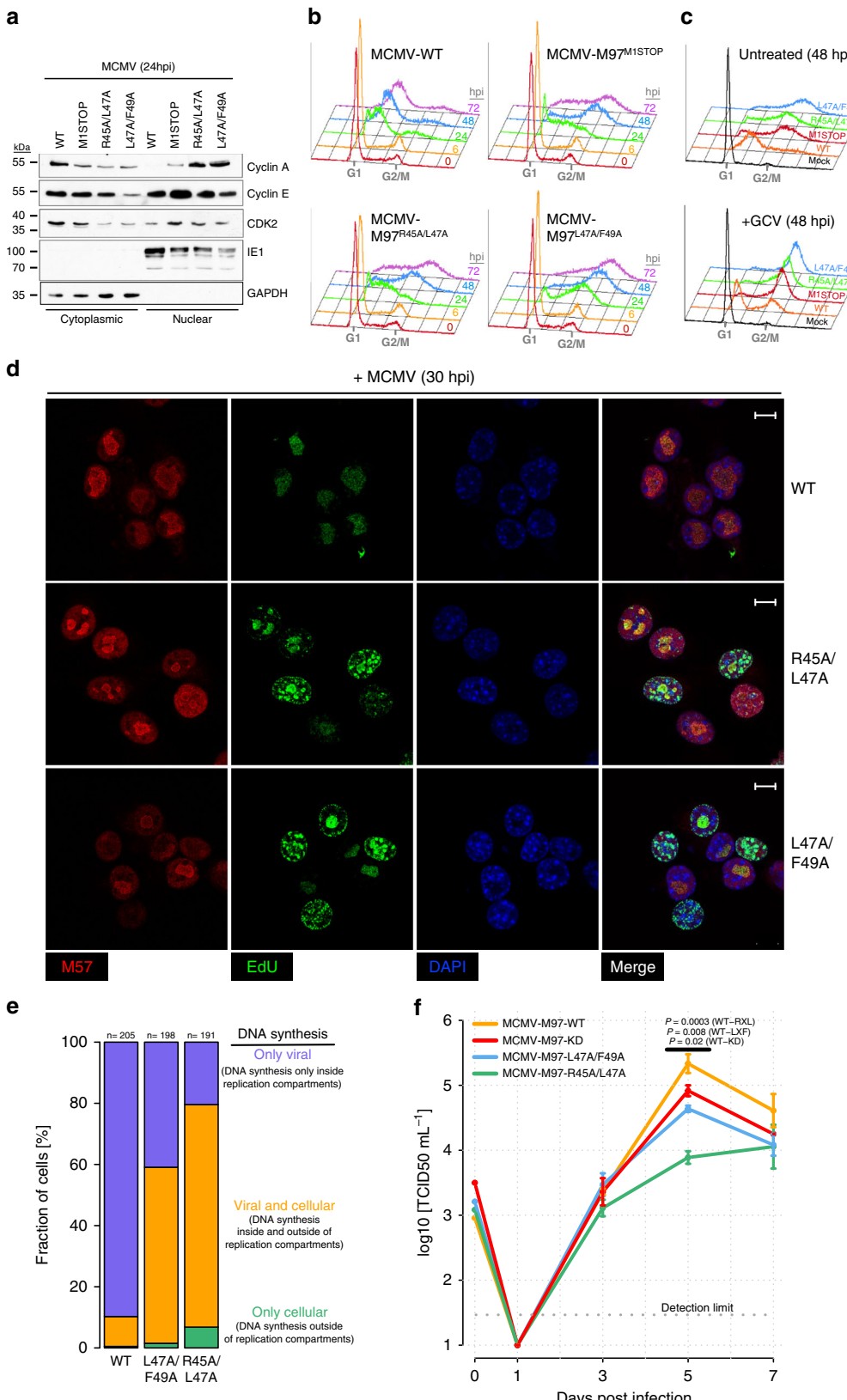

$100\,\mu g\,mL^{-1}$ streptomycin. Where indicated, cells were synchronized in G0/G1 phase by 48 h growth factor deprivation (0.05% serum). In preparation for proteomic analysis, cells were SILAC-labeled for at least five passages using lysine and arginine-deprived DMEM, supplemented with 10% dialyzed serum (cut-off: 10 kDa), 200 mg/L L-proline (only cells destined for phosphoproteome analysis), heavy (L-[$^{13}$C$_6$,$^{15}$N$_2$]-lysine (Lys8), L-[$^{13}$C$_6$,$^{15}$N$_4$]-arginine (Arg10)), medium (L-[$^2$H$_4$]-lysine (Lys4), L-[$^{13}$C$_6$]-arginine (Arg6)) or light (natural lysine (Lys0) and

arginine (Arg0)) amino acids. Labeling efficiency and arginine–proline conversion was checked using LC–MS/MS.

**Viruses**. Viruses were derived from the m129-repaired MCMV strain Smith bacterial artificial chromosome (BAC) pSM3fr-MCK-2fl[66]. Infections were carried out at 37 °C under conditions of centrifugal enhancement. In brief, after a virus

**Fig. 5 M97 causes shutoff of host DNA synthesis by Cyclin A sequestration.** Serum-starved 3T3 cells were infected with the indicated recombinant viruses and subjected to subcellular fractionation (**a**), cell cycle analysis (**b, c**), or confocal microscopy (**d**). **a** The levels of Cyclin A, E, and Cdk2 proteins was determined at 24 h post infection in nuclear and cytosolic fractions by immunoblotting. The soluble viral nuclear protein IE1 and the cytosolic marker GAPDH served as controls. The immunoblots are representative of three independent experiments with similar results. **b, c** The DNA content of infected cells was analyzed by propidium iodide staining followed by flow cytometry and plotted as DNA histograms. **b** The accumulation of viral and cellular DNA was monitored over the time course of infection. **c** To discriminate viral from cellular DNA replication, infected cells were treated with ganciclovir (GCV) or left untreated. **d, e** At 30 hpi, infected cells were pulse-labeled with EdU. EdU staining via click-chemistry (green fluorescence) served to determine sites of cellular and viral DNA synthesis. EdU was combined with immunofluorescence detection of the viral replication factor M57 (red fluorescence) that marks sites of viral DNA synthesis. DAPI was used for nuclear counterstaining (blue fluorescence). Scale bars: 10 μm. **e** The indicated number of cells were categorized based on the (co-)localization of EdU and M57 fluorescence within the nucleus. **f** Mouse embryonic fibroblasts were infected with the indicated viruses at an MOI of 0.02. At the indicated days post infection, the infectious supernatant was harvested and subjected to virus titration. Means (center of the error bars) and standard errors of the mean of $n = 3$ (3, 5, 7 days post infection) biological replicates are depicted. For 0 days post infection: $n = 2$. Two-sided $t$-tests without multiple hypothesis correction were performed comparing the indicated viruses at 5 days post infection.

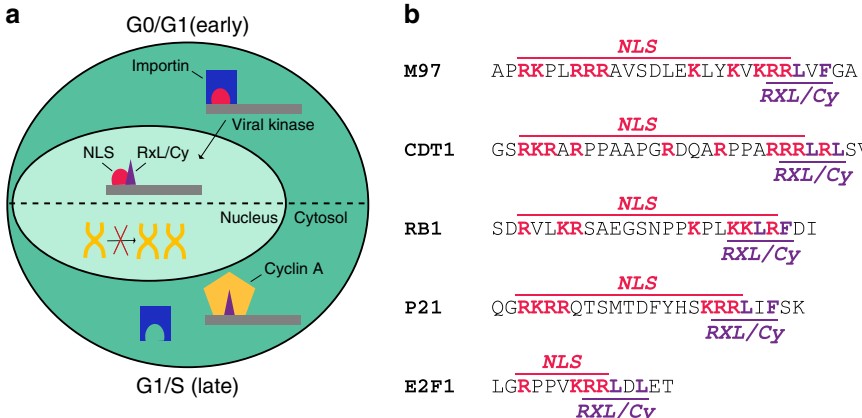

**Fig. 6 The NLS-RXL module is conserved across several cell cycle regulators. a** A model summarizing the function of the NLS-RXL/Cy module in infected cells. In the absence of Cyclin A (G0/G1, early), the NLS of M97 is functional and M97 is imported into the nucleus. MCMV induces Cyclin A and drives the cell cycle towards an S-phase environment (G1/S, late). Cyclin A binds to the RXL/Cy motif on M97 and masks the NLS, leading to cytosolic accumulation of M97–Cyclin A complexes. Cellular DNA synthesis is inhibited due to mislocalized Cyclin A. **b** Conservation of RXL/Cy-NLS modules across several cellular cell cycle regulatory proteins. The depicted sequences are of human origin.

adsorption period of 30 min, cell cultures were centrifuged for 30 min at 1000 $g$. Then, the virus inoculum was replaced by fresh medium. Virus titers were determined by flow cytometry of IE1 fluorescent cells at 6 h post infection or by using the median tissue culture infective dose (TCID$_{50}$) method. Unless otherwise stated, a multiplicity of infection (MOI) of 5 IE protein forming units (IU) per cell was used for experiments.

**Bacmids.** MCMV-HA-M97 and MCMV-M97-K290Q have been described recently[23]. R45A/L47A, L47A/F49A, and M1STOP mutations were introduced into MCMV-HA-M97 by traceless BAC mutagenesis[67]. The oligonucleotide primers used for BAC mutagenesis are specified in Supplementary Data 5. All mutants were controlled by diagnostic PCR and sequencing (Supplementary Fig. 5). To reconstitute infectious virus, BACs together with pp71 expression plasmid were transfected into 3T3 fibroblasts using an Amaxa nucleofector (Lonza).

**Plasmids.** PCGN-based expression plasmids for HA-tagged HHV1-UL13, HHV3-ORF47, HHV4-BGLF4, HHV5-UL97, HHV6-U69, HHV7-U69, and HHV8-ORF36 (Addgene plasmids #26687, #26689, #26691, #26693, #26695, #26697, #26698) were gifts from Robert Kalejta[13]. PCIneo-3HA-M97 was used as expression plasmid for HA-tagged M97 (ref. [23]). RXL/Cy mutations were introduced by site-directed inverse PCR-mutagenesis (primers, see Supplementary Data 5). PHM830 (Addgene plasmid #20702) was a gift from Thomas Stamminger[36]. Fragments encompassing the NLS-RXL/Cy modules of M97 and U69 were PCR-amplified and cloned between NheI and XbaI sites of pHM830 (primers listed in Supplementary Data 5). Plasmids were confirmed by Sanger sequencing and purified by cesium chloride-ethidium bromide equilibrium centrifugation. PEI MAX (Polysciences), transfection grade linear polyethylenimine hydrochloride with a molecular weight of 40 kDa, was used for transfection.

**Phosphopeptide enrichment.** SILAC-labeled cells were harvested 24 h post MCMV infection via scraping. Cells were lysed with 6 M urea/2 M thiourea in 0.1 M Tris-HCl, pH 8.0. Samples were reduced with 10 mM dithiothreitol (DTT) and alkylated with 50 mM iodoacetamide for 30 min in the dark. Proteins were

digested by lysyl endopeptidase (Wako Pure Chemicals) at an enzyme-to-protein ratio of 1:100 (w/w) for 3 h. Subsequently, samples were diluted with 50 mM ammonium bicarbonate to a final concentration of 2 M urea. Digestion with proteomics-grade modified trypsin (Promega) was performed at an enzyme-to-protein ratio of 1:100 (w/w) under constant agitation for 16 h. Enzyme activity was quenched by acidification with trifluoroacetic acid (TFA). The peptides were desalted with C18 Stage Tips prior to nanoLC-MS/MS analysis (whole cell lysate samples). An aliquot of the whole cell lysates were further processed for phosphopeptide enrichment. The tryptic digests corresponding to 200 μg protein per condition were desalted with big C18 Stage Tips packed with 10 mg of ReproSil-Pur 120 C18-AQ 5-μm resin (Dr. Maisch GmbH). Peptides were eluted with 200 μL loading buffer, consisting of 80% acetonitrile (ACN) and 6% TFA (vol/vol). Phosphopeptides were enriched using a microcolumn tip packed with 0.5 mg of TiO$_2$ (Titansphere, GL Sciences). The TiO$_2$ tips were equilibrated with 20 μL of loading buffer via centrifugation at 100 $g$. 50 μL of each sample were loaded on a TiO$_2$ tip via centrifugation at 100 $g$ and this step was repeated until all the sample was loaded. The TiO$_2$ column was washed with 20 μL of the loading buffer, followed by 20 μL of 50% ACN/0.1% TFA (vol/vol). The bound phosphopeptides were eluted using successive elution with 30 μL of 5% ammonium hydroxide and 30 μL of 5% piperidine. Each fraction was collected into a fresh tube containing 30 μL of 20% formic acid. 3 μL of 100% formic acid was added to further acidify the samples. The phosphopeptides were desalted with C18 Stage Tips prior to nanoLC-MS/MS analysis.

**Affinity purification.** At 12 and 36 h post infection (M97 interactome) or 1 day post transfection (HHV kinases interactomes), cells were harvested by scraping in PBS. After centrifugation at 300 $g$, cell pellets were lysed in 25 mM Tris-HCl (pH 7.4), 125 mM NaCl, 1 mM MgCl$_2$, 1% Nonidet P-40, 0.1% sodium dodecyl sulfate (SDS), 5% glycerol, 1 mM DTT, 2 μg mL$^{-1}$ aprotinin, 10 μg mL$^{-1}$ leupeptin, 1 μM pepstatin, 0.1 mM Pefabloc. For HA-AP, a μMACS HA isolation kit (Miltenyi Biotec) was employed according to the manufacturer's instructions, with the following modifications. Lysates were incubated with the magnetic beads for 1 h and SILAC labels were mixed right before the lysates were applied to the column. Lysis buffer was used for the first washing step, lysis buffer without detergent for the

second and 25 mM Tris-HCl (pH 7.4) for the final washing step. Samples were eluted in a total volume of 0.2 mL 8 M guanidine hydrochloride at 95 °C. Proteins were precipitated from the eluates by adding 1.8 mL LiChrosolv ethanol and 1 µL GlycoBlue. After incubation at 4 °C overnight, samples were centrifuged for 1 h at 4 °C and ethanol was decanted before samples were resolved in 6 M urea-2 M thiourea buffer. Finally, samples were reduced, alkylated, digested, and desalted as described above (phosphoproteomics).

**NanoLC-MS/MS analysis.** Phosphopeptides and peptides from whole cell lysates were separated on a MonoCap C18 High Resolution 2000 column (GL Sciences) at a flow rate of 300 nL/min. 6 and 4 h gradients were performed for whole cell peptides and phosphopeptides, respectively. Peptides from HA-AP samples were separated on 45 min, 2 or 4 h gradients with a 250 nL/min flow rate on a 15 cm column (inner diameter: 75 µm), packed in-house with ReproSil-Pur C18-AQ material (Dr. Maisch GmbH). A Q Exactive Plus instrument (Thermo Fisher) was operated in the data-dependent mode with a full scan in the Orbitrap followed by 10 MS/MS scans, using higher-energy collision dissociation. For whole proteome analyses, the full scans were performed with a resolution of 70,000, a target value of $3 \times 10^6$ ions, a maximum injection time of 20 ms and a 2 m/z isolation window. The MS/MS scans were performed with a 17,500 resolution, a $1 \times 10^6$ target value and a 60 ms maximum injection time. For phosphoproteome analysis, the full scans were performed with a resolution of 70,000, a target value of $3 \times 10^6$ ions and a maximum injection time of 120 ms. The MS/MS scans were performed with a 35,000 resolution, a $5 \times 10^5$ target value, 160 ms maximum injection time and a 2 m/z isolation window. For AP–MS of M97 interactomes, full scans were performed at a resolution of 70,000, a target value of $1 \times 10^6$ and maximum injection time 120 ms. MS/MS scans were performed with a resolution of 17,500, a target value of $1 \times 10^5$ and a maximum injection time of 60 ms. Isolation window was set to 4.0 m/z. A Q-Exactive HF-X instrument (Thermo Fisher) or Q-Exactive Plus instrument was used for AP–MS samples of HHV kinases. The Q-Exactive HF-X instrument was run in Top20 data-dependent mode. Full scans were performed at a resolution of 60,000, a target value of $3 \times 10^6$ and maximum injection time of 10 ms. MS/MS scans were performed with a resolution of 15,000, a target value of $1 \times 10^5$ and a maximum injection time of 22 ms. The isolation window was set to 1.3 m/z. The Q-Exactive Plus instrument was run in data-dependent top10 mode. Full scans were performed at a resolution of 70,000 a target value of $3 \times 10^6$ and maximum injection time of 120 ms. MS/MS scans were performed with a resolution of 35,000, a target value of $5 \times 10^5$ and a maximum injection time of 120 ms. The isolation window was set to 4.0 m/z. In all cases normalized collision energy was 26.

**Data analysis.** Raw data were analyzed and processed using MaxQuant 1.5.2.8 (M97 interactomes) or 1.6.0.1 (phosphoproteomics, whole cell lysates, and interactomes of HHV-CHPKs) software[68]. Search parameters included two missed cleavage sites, fixed cysteine carbamidomethyl modification, and variable modifications including methionine oxidation, N-terminal protein acetylation, asparagine–glutamine deamidation. In addition, serine, threonine, and tyrosine phosphorylations were searched as variable modifications for phosphoproteome analysis. Arg10 and Lys8 and Arg6 and Lys4 were set as labels where appropriate. The peptide mass tolerance was 6 ppm for MS scans and 20 ppm for MS/MS scans. The "match between runs" option was disabled and "re-quantify", "iBAQ" (intensity-based absolute quantification) and "second peptide" options were enabled. Database search was performed using Andromeda, the integrated Max-Quant search engine, against a protein database of MCMV strain Smith and a Uniprot database of mus musculus proteins (downloaded July 2015) with common contaminants. Raw data from AP–MS samples of HEK-293T cells were searched against a Uniprot database of human proteins (downloaded August 2018 or October 2016) and the sequences of transgenic HHV kinases including common contaminants. FDR was estimated based on target-decoy competition and set to 1% at PSM, protein and modification site level.

**Bioinformatics of phosphoproteomic profiles.** Phosphosite data and whole proteome data were filtered to exclude contaminants, reverse hits and proteins only identified by site (that is, only identified by a modified peptide). Phosphorylation sites were ranked according to their phosphorylation localization probabilities (P) as class I ($P > 0.75$), class II ($0.75 > P > 0.5$) and class III sites ($P < 0.5$). Class I sites (in at least one of the replicates) were used with a multiplicity of one (that is, only one phosphorylation site on a peptide). MaxQuant normalized site ratios (from Phospho(STY)Sites.txt file) were used and corrected by the ratio of the corresponding protein (from ProteinGroups.txt file) for the respective replicate. SILAC ratios of replicates were log2 transformed, averaged and sites were considered that were quantified in at least one of the replicates. Sites were then categorized as belonging to nuclear or cytosolic proteins, based on the GO annotation of the source protein. Source proteins and corresponding sites with no clear nuclear or cytosolic annotation were classified as "no category". To assess differences in subcellular phosphoproteomic profiles we used the average SILAC ratio of cells infected with R45A/L47A and L47A/F49A mutant viruses and compared phosphosites that belong to nuclear or cytosolic proteins (see above). One-sided Wilcoxon rank-sum test was performed comparing these two subsets of phosphosites with any possible amino acid in the region +4 to −4 (0 refers to the

phosphorylated amino acid). Comparisons were considered when there were at least 19 phosphosites from both cytosolic and nuclear proteins for an amino acid at a given position quantified.

**Bioinformatics of HHV–CHPK interactomes.** AP–MS data were filtered as described above, ratios were log2 transformed and replicates were averaged (mean) when they were quantified in all three replicates. Two-sided one sample $t$-tests (null hypothesis: $\mu_0 = 0$) were performed on the experimental data and a set of simulated data where enrichment ratios were permuted for the individual replicates (999 permutations). The $t$-test $p$-values were then adjusted according to the permuted data. The $p$-values in Supplementary Data 1, Fig.1 and Supplementary Fig. 1 were adjusted in this way. Candidate interactors were selected based on a combination of adjusted $p$-value and means of the three replicates. To harmonize the data obtained from the different CHPK-IPs, we discriminated between candidate interactors and background binders based on volcano plots. For all APs, we used a fixed $p$-value cut-off of 0.05 and a flexible SILAC fold-change cut-off according to an FDR estimation. For this, we used the simulated data as a false positive set and accepted candidate interactors above a SILAC fold-change that recalled maximum 1% false positives.

FDR calculations were based on the simulated data as false positives, as previously suggested[69]. This yielded a set of high-confidence candidate interactors for APs with individual HHV kinases. To compare individual prey proteins across the APs with different kinases we imputed missing values with random values from a normal distribution (with mean 0 and standard deviation 0.25). Enrichment profiles were clustered using Euclidean distance and assembled into a heatmap using R. GO enrichment of the prey proteins in selected sets of clusters were performed using Metascape[70].

**Bioinformatics of M97 interactomes.** AP–MS data were filtered as described above, ratios were log2 transformed and replicates were averaged (mean) when they were quantified in at least two of the replicates. Two-sided one sample $t$-tests (null hypothesis: $\mu_0 = 0$) were performed on the experimental data and proteins were considered as interactors when they were below a $t$-test $p$-value of 0.05 and above a log2 SILAC fold-change of 1.5. Additionally, the molar amount of bait and prey proteins in MS samples was estimated by iBAQ values. Therefore, for samples where M97 was purified, the iBAQ values were summed up, sorted and log10 transformed.

**Immunoblot analysis.** Whole cells were harvested and subsequently lysed by sonication in 50 mM Tris-Cl (pH 6.8), 2% SDS, 10% glycerol, 1 mM DTT, 2 µg mL$^{-1}$ aprotinin, 10 µg mL$^{-1}$ leupeptin, 1 µM pepstatin, 0.1 mM Pefabloc. The lysates were clarified by centrifugation and protein concentration was measured using the Bio-Rad DC protein assay. Lysates were adjusted to equal protein concentration, supplemented with 100 mM dithiothreitol and bromophenol blue, and boiled at 95 °C for 3 min. For subcellular fractionation into nuclei and cytoplasmic extracts, cells were lysed by Dounce homogenization in hypotonic buffer, consisting of 10 mM Hepes (pH 8.0), 10 mM KCl, 1.5 mM MgCl$_2$, 0.34 M sucrose, 10% glycerol, 0.1 mM DTT and protease inhibitors. Nuclei were collected by low-speed centrifugation (4 min, 1300 $g$, 4 °C). The supernatant was clarified by high-speed centrifugation (15 min, 20,000 $g$, 4 °C). The nuclei were washed once in the hypotonic extraction buffer and lysed then as described above for the preparation of whole cell lysates. Proteins were resolved by SDS-polyacrylamide gel electrophoresis (PAGE) and blotted to polyvinylidene fluoride membranes. To prevent nonspecific binding, blots were incubated in Tris-buffered saline-0.1% Tween-20 (TTBS) supplemented with 5% skim milk. Afterwards blots were incubated with the following primary antibodies: Cyclin A, C19 (Santa Cruz); cyclin E, M20 (Santa Cruz); CDK2, M2 (Santa Cruz); HA, clone 3F10 (Roche); M99, mouse antiserum (generously provided by Khanh Le-Trilling, University Hospital Essen); IE1, clone Croma 101; M45, clone M45.01; M57, clone M57.02 (all obtained from Center for Proteomics, Rijeka). The blots were developed using horseradish peroxidase-conjugated secondary antibodies in conjunction with the Super Signal West Dura chemiluminescence detection system (Thermo Fisher). All antibodies were diluted to 1 µg per mL in TTBS, 5% skim milk. Uncropped scans of the immunoblots are provided in the Source Data file.

**Immunoprecipitation.** Cells were lysed by freezing-thawing in IP buffer (IPB): 50 mM Tris-Cl pH 7.4, 150 mM NaCl, 10 mM MgCl$_2$, 10 mM NaF, 0.5 mM Na$_3$VO$_4$, 0.5% Nonidet P-40, 10% glycerol, 1 mM DTT, 2 µg mL$^{-1}$ aprotinin, 1 mM leupeptin, 1 mM Pefabloc. Cell extracts were clarified by centrifugation at 20,000 $g$. Cyclin A IPs were performed by incubating the IPB extracts with Cyclin A, H432, conjugated agarose beads (Santa Cruz)[46]. For HA IPs, a µMACS HA isolation kit (Miltenyi Biotec) was employed according to the manufacturer's instructions, except that IPB was used as both lysis and washing buffer.

**Kinase assay.** First, HA-M97 was immunoprecipitated from infected cells. To this end, IPB extracts were prepared and incubated with HA antibody clone 3F10 (1 µg per mL) and Protein G-conjugated agarose beads. The precipitates were washed several times with IPB and twice with 20 mM Tris-Cl (pH 7.4), 10 mM MgCl$_2$, 1 mM DTT. Then, they were incubated under constant agitation for 60 min at 30 °C in kinase reaction buffer: 20 mM Tris-HCl (pH 7.4), 10 mM MgCl$_2$, 1 mM DTT,

10 mM β-glycerophosphate, 50 μM ATP, 5 μCi [γ-$^{32}$P]ATP. Kinase reactions were analyzed by 8% SDS-PAGE followed by autoradiography.

**Immunofluorescence microscopy.** 3T3 fibroblasts were grown and infected on glass coverslips. Where indicated, cells were incubated with 10 μM 5-ethynyl-2′-deoxyuridine (EdU) for 30 min. To analyze for M97 localization and sites of EdU incorporation, coverslips were washed with PBS and incubated for 10 min in PBS-4% paraformaldehyde fixation solution, followed by additional washing and incubation in PBS-T permeabilization solution (PBS, 0.1% Triton X-100, 0.05% Tween 20) and 2% bovine serum albumin (BSA) fraction V/ PBS-T blocking solution. Afterwards, incorporated EdU was conjugated to Alexa Fluor 488 using the Click-iT EdU labeling kit (Thermo Fisher). Then samples were incubated overnight at 4 °C with the following antibodies: anti-HA clone 3F10 or anti-M57 clone M57.02 (Center for Proteomics, Rijeka), both diluted to 1 μg mL$^{-1}$ in PBS-T containing 2% BSA. After washing in PBS, cells were incubated for 1 h at 25 °C with Alexa Fluor 488 or 647-coupled anti-IgG antibodies (Thermo Fisher). Coverslips were mounted on glass slides in 4′,6-diamidino-2-phenylindole (DAPI) containing Fluoromount-G medium (Thermo Fisher). Images were acquired by an Eclipse A1 laser-scanning microscope, using NIS-Elements software (Nikon Instruments). Equal microscope settings and exposure times were used to allow direct comparison between samples. For quantification, the microscope slides were randomly scanned and all cells in the randomly acquired frames were analyzed by ImageJ.

**Flow cytometry.** Cells were fixed and permeabilized in ice-cold PBS-80% ethanol for at least 16 h. After washing, cells were incubated on ice for at least 16 h with one of the following primary antibodies: anti-IE1 (clone Croma 101) or anti M57 (clone m57.02), both diluted to 1 μg mL$^{-1}$ in PBS-1% BSA. After washing, cells were incubated in Alexa Fluor 647 conjugated anti-mouse IgG for 1 h at 25 °C. Cells were washed again and incubated then for 15 min at 25 °C in PBS supplemented with 0.1 mg mL$^{-1}$ RNAse A and 25 μg mL$^{-1}$ propidium iodide. All washing steps and antibody dilution were performed using PBS-1% BSA. Flow cytometry was performed using a FACSCanto II instrument equipped with FACSDiva and Cell-Quest Pro software (Becton Dickinson).

**Reporting summary.** Further information on research design is available in the Nature Research Reporting Summary linked to this article.

## Data availability

MS raw data and MaxQuant output tables have been deposited to the ProteomeXchange Consortium via the PRIDE partner repository with the dataset identifier via PXD016334. URL. The source data underlying Figs. 1e, 2b, e, 3b–f, 4b, 5a, e, f and Supplementary Figs. 1a, b, 3c–f and 10b are provided as a Source Data file. The source data underlying Figs. 1b–d, 3a, 4c and Supplementary Figs. 1a, c, 3b, 6b–e, 8b, c are provided in Supplementary Data files 1–4. Fasta files for proteomic searches were downloaded from Uniprot (https://www.uniprot.org/). All other relevant data are available from the corresponding author upon reasonable request. Source data are provided with this paper.

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

## Acknowledgements

The authors thank Jens von Einem, Tihana Lenac, Khan-Le Trilling, and Matthias Truss for sharing valuable reagents. This work was supported by a grant (#900005) from the Joachim-Herz-Stiftung to L.W. M.Sc. was funded by a MD student research stipend from the Berlin Institute of Health.

## Author contributions

Conceptualization, B.B., M.Sc., L.W.; Methodology, B.B., M.Sc., H.W., K.I., E.O., M.Se., L.W.; Formal analysis, B.B., H.W., E.O., L.W.; Investigation, B.B., M.Sc., H.W., I.G., K.I., E.O., L.W.; Resources, M.Sc., I.G., B.V., M.Se., L.W.; Data curation, B.B.; Visualization, B.B., L.W.; Writing—original draft, B.B., M.Sc., L.W.; Writing—review and editing, B.B., L.W.; Funding acquisition, M.Sc., W.B., M.Se., C.H., L.W.; Supervision, W.B., M.Se., C.H., L.W.

## Funding

## Competing interests

The authors declare no competing interests.
