## [Peer Review File · Nature Communications]

Reviewers' comments:

Reviewer #1 (Remarks to the Author):

In this study, Bogdanow and colleagues utilized quantitative proteomic approaches to identify cellular proteins that interact with the conserved herpesvirus protein kinases (CHPKs). Globally, the authors found that CHPKs target key regulators in the categories of transcription and replication. The authors further focused on the mechanistic targeting of Cyclin A and associated factors by CHPKs encoded by beta-herpesvirus. They demonstrated that beta-herpesvirus CHPKs RXL/Cy motif is required for Cyclin A recruitment, that Cyclin A binding inhibits the function of the adjacent nuclear localization signals (NLS) within viral kinases of HHV6/7 and cytomegaloviruses (CMVs), and consequently, that the kinase localization and substrate targeting are also altered. On the other hand, the authors demonstrated that viral kinases redirect Cyclin A to the cytosol to inhibit cellular DNA replication upon viral infection. Overall this study provides a valuable resource for the community to further delineate the substrates targeting by the CHPKs.

Major points,

The authors used HA-tagged CHPKs to pull down cellular proteins associated with viral kinases. One caveat is that the expression levels of these kinases were not shown by western blot. According to the publications by Kuny et al (PLOS Path, 2010) and Zhang et al (Cell Rep, 2019), the kinase expression levels varied significantly across different CHPKs. If this is the case for this study, then the binding proteins may differ due to the kinase expression levels. Therefore the authors should take this into consideration for comparing common/different binders across different CHPKs.

Minor Points,

1. Figure 1. Although human CMV kinase pUL97 can pull down Cyclin A and CDK2 (Fig. 1d), the subsequent Co-IP failed to detect such interaction (Fig. 1e).
2. Figure 3e, the asterisk label is not defined in the text. For HA-M97 IP results, it is not clear why the WT protein expression is higher in the input while after HA-IP there is significant less protein compared to 3 other mutants. Is this amount of protein used for in vitro auto-phosphorylation assay? If it is the case, the mutant proteins are less active compared to WT.
3. Page 8, line 204. The text reads as "...at 6 h post infection" but in figure 4a, it is labeled as 8 hpi.
4. Page 9, line 213. Fig 3a should be "Fig 4a"
5. Page 11, line 267. The text reads as "...show a reduced size of viral replication compartments" while I noticed that M57 signals in R45A/L47A are similar or stronger than WT (Fig 5d).

6. Figure 6b. It is not clear whether the cellular protein sequences are from human or other species. Since the authors are comparing MCMV kinase M97, it will be informative to also include mouse protein sequences here.

Reviewer #2 (Remarks to the Author):

Work presented here shows that beta-herpesvirus kinases utilize an overlapping nuclear localization signal (NLS) and cyclin-binding site (RxL) to control kinase localization, substrate phosphorylation, and the inhibition of cellular DNA synthesis. When cyclin A levels are low, the NLS directs the viral kinase to the nucleus where nuclear proteins are phosphorylated. As cyclin A levels rise, cyclin A associates with the kinase, blocks the NLS, causes an accumulation of the kinase in the cytoplasm, and thus the phosphorylation of cytoplasmic substrates. Importantly, cyclin A sequestration in the cytoplasm by the kinase inhibits cellular DNA synthesis. This is a unique and interesting regulatory system that may function for cellular proteins as well. The findings, along with a wealth of viral kinase interactome data, represent a significant contribution to the field. Some minor additions would improve the manuscript, as listed below.

1. The experiments in Figure 2d would benefit from quantitation and statistical analysis.
2. The MuvB complex (line 186) is also regulated by HCMV UL97 (Iwahori & Kalejta 2017 Virology 512:95-103)
3. The v-Cdks have been described as “cyclin-independent” (Ref 19) (Line 197).
4. The experiments in Figure 4a (wrong citation in line 213?) would benefit from corroboration in cyclin A-deficient (knockdown or knockout) cells.
5. The major element missing from the manuscript is discussion or demonstration of what this system means for viral infection in vitro or pathogenesis in vivo. I think it would be sufficient to speculate about in vivo pathogenesis. But it seems reasonable to know the phenotypes (growth curves) of the mutant viruses used throughout the paper (M1STOP, R45A/L47A, L47A/F49A), and (perhaps) whether those phenotypes were affected by the absence of cyclin A.

Reviewer #3 (Remarks to the Author):

Bogdanov and colleagues determined the interactome of seven herpesvirus kinases by quantitative AP-MS. Cyclin A is identified and validated as a specific interactor of the beta-herpesvirus family. Interaction is shown to involve a docking interaction with the (unstructured) N-terminal tail of the kinase and overlapping with a bipartite nuclear localisation signal (NLS). The authors present convincing data using mutated viruses with point mutation in the cyclin docking motif and NLS to demonstrate that cyclin binding masks the NLS and results in higher levels of cyclin A in the cytosol and results in inhibition of DNA replication during infection.

This is an interesting manuscript that has systematically studied the interactome of important viral proteins and derived a molecular mechanism that is possibly important for the herpesvirus infection cycle. The technical quality of the manuscript is good. The experiments are generally well controlled and appropriately interpreted.

The described mechanism of cyclin binding blocking an NLS is not entirely novel and there are dozens of other examples in which binding of a protein-protein interaction domain or a PTM interferes with the function of an adjacent short-linear motifs. In addition, there are a few major points that should be addressed before the manuscript can be considered as a strong candidate for Nature Communications.

Major points:

1. It is regrettable that the AP-MS dataset of the selected CHPKs is not validated in a bit more detail. For example, the strong interaction of HHV5-UL97 with SUMO/PRC1 proteins and of HHV4-BGLF4 with chromatin silencing/DNA replication protein (see heatmap Fig 1.c), which appears to be strikingly selective for the respective CHPKs would merit validation by IP or ChIP (in parallel for the binding and non-binding CHPK controls). This would increase my confidence and trust in the AP-MS dataset.
2. Fig 1e: It appears that the overexpression levels of the HA-tagged kinases are dramatically different (possibly up to 100-fold). I suppose the same is true for the extracts that were used for the SILAC experiment (Fig 1a-c). This raises concerns about possible artefacts caused by heavy overexpression and problems to compare interactors across experiments. An HA-immunoblot of the extracts that were used for the AP-MS experiments should be added to SI. Likewise, an immunoblot experiment that demonstrates the efficiency of the HA pull-down for the different HA-tagged kinases should be shown and a silver gel of a fraction of the eluates (or another method) to judge the complexity/concentration of the protein complexes that were pulled down should be provided with a revised version of the manuscript.

3. On the same point, were there attempts to determine bait protein expression levels by MS and to normalize for the different HA-tagged kinase expression levels. For direct and reliable comparisons of different kinases, two HA-tagged kinases should be compared in the same experiment (e.g. kinase 1: heavy, kinase 2: light SILAC label).
4. Given the proposed interaction mode of the cyclin A with certain CHPKs, it is surprising that human CDK1 and CDK2 were identified as specific interactors as well. What is this explanation? Is there a ternary complex of CHPK-cyclinA-CDK1/2 or is cyclinA binding to a CHPK mutually exclusive to binding to CDK1/2?
5. lines 167-176: The conclusions from the experiments with the predicted NLS motif transplanted into a reporter construct (Fig 2 c-d) should be toned down: NLSs can be regulated in multiple ways and the presented data is at best indicative of a functional NLS. It is by no means 'confirming' (line 171), in particular as the RXL to AXA mutation only leads to a partial phenotype in HHV5-U69.
6. All immunoblot experiments and gels should contain molecular weight markers.
7. For the immunofluorescence experiments, the authors should add statements how cells were selected and if the conclusions are only based on the few cells shown in the figure panels or if quantification was done.

Point by point response

We thank all three reviewers for carefully reading our manuscript and for their constructive and helpful comments! We were pleased to read that we provide a “valuable resource” (reviewer #1), that our study is “interesting”, “appropriately interpreted” and “well-controlled” (reviewer #3) and that it “represent[s] a significant contribution to the field” (reviewer #2).

We carefully addressed the points raised by the reviewers and performed a number of additional experiments/ analyses that strengthen the manuscript and support its conclusions:

- 1. We provide growth curves of cyclin binding deficient mutant viruses in primary cells (**Fig. 5f**). This indicates that cyclin binding is indeed important for efficient replication of MCMV.*
- 2. We generated Cyclin A deficient murine fibroblasts and were able to show that Cyclin A is in fact the crucial target for the M97-induced shutdown of cellular DNA replication (**Supplementary Fig. 10**).*
- 3. We validated additional four new interactors of the AP-MS dataset (**Supplementary Fig. 3c-f**). Among them are proteins involved in transcription, which supports our finding that CHPKs target master regulators of transcription.*
- 4. We performed AP-MS experiments directly comparing APs of HA-pUL97 and HA-BGLF4 (**Supplementary Fig. 3a,b**). In consistency with our original data, we observed a number of interactors that are very specific to the individual kinase. This corroborates our conclusion that there is divergence in the interactomes of different CHPKs.*
- 5. We provide further quality control experiments on the AP-MS dataset, including an analysis of bait expression levels and sample complexity (**Supplementary Fig. 1**).*
- 6. We added quantitative data to all microscopic experiments (**Fig. 2e, 4b, 5e**).*

Below we provide a detailed point by point response. We hope that our manuscript is now suitable for publication in Nature Communications.

Reviewer #1:

In this study, Bogdanow and colleagues utilized quantitative proteomic approaches to identify cellular proteins that interact with the conserved herpesvirus protein kinases (CHPKs). Globally, the authors found that CHPKs target key regulators in the categories of transcription and replication. The authors further focused on the mechanistic targeting of Cyclin A and associated factors by CHPKs encoded by beta-herpesvirus. They demonstrated that beta-herpesvirus CHPKs RXL/Cy motif is required for Cyclin A recruitment, that Cyclin A binding inhibits the function of the adjacent nuclear localization signals (NLS) within viral kinases of HHV6/7 and cytomegaloviruses (CMVs), and consequently, that the kinase localization and substrate targeting are also altered. On the other hand, the authors demonstrated that viral kinases redirect Cyclin A to the cytosol to inhibit cellular DNA replication upon viral infection. Overall this study provides a valuable resource for the community to further delineate the substrates targeting by the CHPKs.

We thank the reviewer for the encouraging feedback!

Major points,

The authors used HA-tagged CHPKs to pull down cellular proteins associated with viral kinases. One caveat is that the expression levels of these kinases were not shown by western blot. According to the publications by Kuny et al (PLOS Path, 2010) and Zhang et al (Cell Rep, 2019), the kinase expression levels varied significantly across different CHPKs. If this is the case for this study, then the binding proteins may differ due to the kinase expression levels. Therefore the authors should take this into consideration for comparing common/different binders across different CHPKs.

*The reviewer is right. In consistency with the data from Kuny et al., and Zhang et al., we also observed that kinase expression levels varied significantly when overexpressed. Following the reviewers comment, we now provide an HA-Immunoblot of cell lysates transfected with the individual kinases in our **new Supplementary Fig. 1a** (see our revised manuscript). This shows that the abundance of the tagged kinases varies by about ~70-80 fold at the extremes. We observed the biggest differences in abundance between the U69 kinases of HHV6 and HHV7. These two kinases share also the largest set of interactors, suggesting that our proteomic approach is robust and not so sensitive towards differences in overall bait expression. In addition, despite the vast differences in bait protein abundance, the cyclin A2-CDK complex and associated factors co-enriched with HHV6/7-CHPKs to a similar extent, as indicated by the log2 fold-changes (**Fig. 1d**).*

To directly test whether the protein abundance of the bait in the cell extract has a major influence on the identified interaction partners we performed an experiment where we transfected expression plasmids for U69 of HHV6 in two different amounts (2.5 µg and 25 µg) and subjected the samples to shotgun proteomics. Again, we compared enriched proteins to an internal control that was empty vector transfected and performed the experiment in duplicates (see **Fig.R1** below). Even though the U69 expression levels in the cell extracts used as AP-MS input varied significantly (**Fig. R1a**), nicely reflecting the two different transfection conditions, we retrieved almost the same set of protein interactors for both conditions (**Fig. R1b**). In consistency with our original data, we observed co-enrichment of the chaperonin containing TCP1 (CCT) complex as well as cyclins A/B and CDK1/2 with both conditions. Thus, transfecting 10x lower amounts of expression plasmid had a rather minor impact on the retrieved proteins.

In general, it is of course possible that differences in bait protein abundance in the cell extract can lead to differences in the detected interaction partners. To acknowledge that protein abundance may perhaps alter the interacting proteins we added a sentence into the existing paragraph (last paragraph in the Discussion section) discussing general problems when using transfection approaches for interactome studies.

“It is important to note that this type of experiment cannot resolve protein interactions that depend on the environment of an infected cell. For example, the CHPK may be differentially abundant, differentially modified, differentially folded or differentially localized in a transfected compared to an infected cell. It is also critical to interpret our findings in the broader context of kinase-associated functions that dynamically change during infection.”

Please see also our response to reviewer #3, point 2 and 3.

Fig. R1: Differences in the interactome HA-U69(HHV6) when changing the amount of transfected plasmid 10-fold. (a) anti-HA and anti-GAPDH (loading control) immunoblots of lysates from SILAC cells transfected with the indicated amount of expression vectors. **(b)** Lysates were used for HA-AP MS experiments. The experiment was designed in label-swap duplicates and the average SILAC fold-changes are depicted.

Minor Points,

1. Figure 1. Although human CMV kinase pUL97 can pull down Cyclin A and CDK2 (Fig. 1d), the subsequent Co-IP failed to detect such interaction (Fig. 1e).

We agree that the pUL97 western blot band in the output of the Cyclin A co-IP experiment is not easily visible. We now include a longer HA western blot exposure in Fig. 1e, which improves the visibility of the HA-UL97 band in cyclin A co-immunoprecipitates.

2. Figure 3e, the asterisk label is not defined in the text. For HA-M97 IP results, it is not clear why the WT protein expression is higher in the input while after HA-IP there is significantly less protein compared to 3 other mutants. Is this amount of protein used for in vitro auto-phosphorylation assay? If it is the case, the mutant proteins are less active compared to WT.

We thank the reviewer for this careful comment. The asterisk denotes a non-specific band of a protein in the lysate. The input used for in-vitro autophosphorylation is the HA-IP. The apparently lower levels of WT are due to a blotting artifact. Therefore, we now include a different replicate of the same experiment in Fig. 3e in our revised version. In order not to confuse the readers, we now include only the input that was used for the HA-IP.

3. Page 8, line 204. The text reads as "...at 6 h post infection" but in figure 4a, it is labeled as 8 hpi. Page 9, line 213. Fig 3a should be "Fig 4a".

We have corrected these points in the revised manuscript.

4. Page 11, line 267. The text reads as "...show a reduced size of viral replication compartments" while I noticed that M57 signals in R45A/L47A are similar or stronger than WT (Fig 5d).

As this observation is not important for the overall conclusions, we decided to remove the corresponding sentence from the manuscript.

5. Figure 6b. It is not clear whether the cellular protein sequences are from human or other species. Since the authors are comparing MCMV kinase M97, it will be informative to also include mouse protein sequences here.

*The cellular protein sequences shown in Fig. 6b are of human origin. We now point to this fact in the figure legend. We agree that it would be important to know if overlapping NLS-RxL/Cy sequence motifs in some key cell cycle regulators are conserved in other species. We therefore generated multiple sequence alignments using ProViz and present the data as **new Supplementary Fig. 12**. It turns out that the composite NLS-RxL/Cy sequences are highly conserved throughout vertebrate orthologues, indicating that it is a functionally relevant feature.*

Reviewer #2:

Work presented here shows that beta-herpesvirus kinases utilize an overlapping nuclear localization signal (NLS) and cyclin-binding site (RXL) to control kinase localization, substrate phosphorylation, and the inhibition of cellular DNA synthesis. When cyclin A levels are low, the NLS directs the viral kinase to the nucleus where nuclear proteins are phosphorylated. As cyclin A levels rise, cyclin A associates with the kinase, blocks the NLS, causes an accumulation of the kinase in the cytoplasm, and thus the phosphorylation of cytoplasmic substrates. Importantly, cyclin A sequestration in the cytoplasm by the kinase inhibits cellular DNA synthesis. This is a unique and interesting regulatory system that may function for cellular proteins as well. The findings, along with a wealth of viral kinase interactome data, represent a significant contribution to the field. Some minor additions would improve the manuscript, as listed below.

We thank the reviewer for the positive feedback!

1. The experiments in Figure 2d would benefit from quantitation and statistical analysis.

Following the reviewers comment, we now provide quantitative data for all microscopic experiments that were performed throughout the manuscript (Fig. 2e, Fig. 4b, Fig. 5e).

2. The MuvB complex (line 186) is also regulated by HCMV UL97 (Iwahori & Kalejta 2017 Virology 512:95-103)

We integrated the requested reference (# 44) at the position in the text and rephrased the corresponding sentence as:

“In the late phase (36 h), additional viral and cellular factors co-purified with M97 (Supplementary Fig. 4d). This includes M50-M53, the nuclear egress complex of MCMV⁴², and Lin54, the DNA-binding subunit of the cell cycle-regulatory MuvB complex⁴³, which is known to be regulated by pUL97⁴⁴.”

3. The v-Cdks have been described as “cyclin-independent” (Ref 19) (Line 197).

We thank the reviewer for this thoughtful comment and included this reference at the suggested position in the text:

“Thus, M97-Cyclin A binding has no influence on abundance and enzymatic activity of the involved interaction partners. This is consistent with the view that β -herpesviral CHPKs are cyclin-independent kinases¹⁹.”

4. The experiments in Figure 4a (wrong citation in line 213?) would benefit from corroboration in cyclin A-deficient (knockdown or knockout) cells.

*We thank the reviewer for this excellent idea. Indeed, it would be interesting to see what happens when cyclin A is absent from the cells. Following this idea, we created cyclin A knock-down cell lines (see **new Supplementary Fig. 10a-b**). We obtained very good knock-down efficiency by stable expression of a Cyclin A-specific shRNA (sh5) previously validated by others.*

*Instead of investigating the Cyclin A-dependency of M97 re-localization at late times of infection, we chose to assess cell cycle progression as the most important functional read-out (see the title of the manuscript: “Cross-regulation of viral kinases with cyclin A **secures shutoff of host DNA synthesis**”).*

*As expected, in scrambled shRNA or empty vector transduced control cells, we observed Cyclin A induction by all MCMV-M97 variants (**Supplementary Fig. 10b**) and induction of cellular DNA synthesis and S/G2 cell cycle progression in the absence of M97-Cyclin A binding (**Supplementary Fig. 10c**, M97-R45A/L47A and L47A/F49A mutants). In contrast, in sh5-transduced cells, not only the induction of Cyclin A was successfully prevented (**Supplementary Fig. 10b**) but also the G1 arrest was re-installed in the majority of M97-R45A/L47A and L47A/F49A infected cells.*

Thus, the main cause of uncontrolled cell cycle progression in R45A/L47A or L47A/F49A mutant virus infected cells is the loss of Cyclin A binding and not other up-regulated cyclins like Cyclin E or B.

5. The major element missing from the manuscript is discussion or demonstration of what this system means for viral infection in vitro or pathogenesis in vivo. I think it would be sufficient to speculate about in vivo pathogenesis. But it seems reasonable to know the phenotypes (growth curves) of the mutant viruses used throughout the paper (M1STOP, R45A/L47A, L47A/F49A), and (perhaps) whether those phenotypes were affected by the absence of cyclin A.

*We thank the reviewer for the suggestion. We performed the suggested analysis in our revised version of the manuscript (see **Figure 5f**) for the wildtype and three mutant viruses (R45A/L47A, L47A/F49A, KD). This identifies a significant defect in titers for the mutant viruses compared to wildtype, especially at 5 dpi. The growth defect was more pronounced for the R45A/L47A and L47A/F49A mutant viruses compared to the kinase dead mutant. This indicates that destruction of cyclin binding of the viral kinase is detrimental for viral replication beyond what is seen when the kinase is inactive. While we think that*

it is also a good idea to test the growth of mutant and wildtype viruses under absence of cyclin A, we feel that is beyond the scope of the manuscript (which is already quite long).

Reviewer #3:

Bogdanow and colleagues determined the interactome of seven herpesvirus kinases by quantitative AP-MS. Cyclin A is identified and validated as a specific interactor of the beta-herpesvirus family. Interaction is shown to involve a docking interaction with the (unstructured) N-terminal tail of the kinase and overlapping with a bipartite nuclear localisation signal (NLS). The authors present convincing data using mutated viruses with point mutation in the cyclin docking motif and NLS to demonstrate that cyclin binding masks the NLS and results in higher levels of cyclin A in the cytosol and results in inhibition of DNA replication during infection. This is an interesting manuscript that has systematically studied the interactome of important viral proteins and derived a molecular mechanism that is possibly important for the herpesvirus infection cycle. The technical quality of the manuscript is good. The experiments are generally well controlled and appropriately interpreted.

We thank the reviewer for carefully reading our manuscript and for his/ her constructive feedback.

The described mechanism of cyclin binding blocking an NLS is not entirely novel and there are dozens of other examples in which binding of a protein-protein interaction domain or a PTM interferes with the function of an adjacent short-linear motifs. In addition, there are a few major points that should be addressed before the manuscript can be considered as a strong candidate for Nature Communications.

To the best of our knowledge this is the first report of cyclin binding blocking an NLS. We agree that there are other examples where a PTM or another protein interaction impairs the function of an adjacent SLiM. Importantly, we do not claim that the presented blocking of the NLS by cyclin A is the first description of an NLS being blocked by a binding protein. Instead, we discuss this in the manuscript and cite several other examples:

“This puts β -herpesvirus kinases in a row with a number of cellular and viral proteins known to control nucleo-cytoplasmic localization via intermolecular NLS-masking⁵⁹, with NF- κ B as the best understood example^{60,61}.”

Major points:

1. It is regrettable that the AP-MS dataset of the selected CHPKs is not validated in a bit more detail. For example, the strong interaction of HHV5-UL97 with SUMO/PRC1 proteins and of HHV4-BGLF4 with chromatin silencing/DNA replication protein (see heat map Fig 1.c), which appears to be strikingly selective for the respective CHPKs would merit validation by IP or ChIP (in parallel for the binding and non-binding CHPK controls). This would increase my confidence and trust in the AP-MS dataset.

*We thank the reviewer for his/ her suggestion. Following the comment we validated four novel interactions that were reported in our manuscript (see **new Supplementary Fig. 3c-f**). This validation confirmed the UL97 interactors PRC1 (BMI1), TRIM28 and CCAR2. All these proteins are involved in transcriptional repression.*

2. Fig 1e: It appears that the overexpression levels of the HA-tagged kinases are dramatically different (possibly up to 100-fold). I suppose the same is true for the extracts that were used for the SILAC experiment (Fig 1a-c). This raises concerns about possible artefacts caused by heavy overexpression and problems to compare interactors across experiments. An HA-immunoblot of the extracts that were used for the AP-MS experiments should be added to SI. Likewise, an immunoblot experiment that demonstrates the efficiency of the HA pull-down for the different HA-tagged kinases should be shown and a silver gel of a fraction of the eluates (or another method) to judge the complexity/concentration of the protein complexes that were pulled down should be provided with a revised version of the manuscript.

*The reviewer is right. As also outlined to reviewer #1, point 1, we observed that kinase expression levels varied significantly when overexpressed. Following the reviewers comment, we now provide the requested HA-Immunoblot of cell lysates transfected with the individual kinases in our **new Supplementary Fig. 1a**. This shows that the abundance of the tagged kinases varies by about ~70-80 fold at the extremes. We observed the biggest differences in abundance between the U69 kinases of HHV6 and HHV7. Importantly, these two kinases also share the largest set of interactors, suggesting that our proteomic approach is robust and not so sensitive towards differences in overall bait levels. In addition, despite the vast differences in bait protein abundance, the cyclin A-CDK complex and associated factors co-enriched with HHV6/7-CHKs to a similar extent, as indicated by the log₂ fold-changes (**Fig. 1d**).*

*To directly test whether the protein abundance of the bait in the cell extract has a major influence on the identified interaction partners we performed an experiment where we transfected expression plasmids for U69 of HHV6 in two different amounts (2.5 µg and 25 µg) and subjected the samples to shotgun proteomics. Again, we compared enriched proteins to an internal control that was empty vector transfected and performed the experiment in duplicates (see **Fig.R1** below). Even though the U69 expression levels in the cell extracts used as AP-MS input varied significantly (**Fig. R1a**), nicely reflecting the two different transfection conditions, we retrieved almost the same set of protein interactors for both conditions (**Fig. R1b**). In consistency with our original data, we observed co-enrichment of the chaperonin containing TCP1 (CCT) complex as well as cyclins A/B and CDK1/2 with*

both conditions. Thus, transfecting 10x lower amounts of expression plasmid had a rather minor impact on the retrieved proteins.

We further tested the efficiency of immuno-enrichment of the individual HA-tagged kinases as the reviewer suggested. Instead of using an HA pulldown we decided to use HA-affinity purification (anti-HA μ MACS system by Miltenyi), the same method we employed for enrichment of bait-associated protein complexes in our AP-MS setup. An HA-immunoblot of the input material and the eluates from the μ MACS columns is depicted in **Supplementary Fig. 1b**. In general, the kinases in the input were as abundant as in the eluates, relative to each other. An exception is HHV6-U69 which is underrepresented possibly due to saturation effects caused by limiting amounts of anti-HA conjugated microbeads.

To directly estimate bait abundance in the MS samples, we added an additional plot as **new Supplementary Fig. 1c** where proteins were quantified by iBAQ values. This shows that the bait proteins were typically among the 10 most abundant proteins in the individual eluates (exception: HHV7-U69, rep1: rank 27, rep2: rank 33, rep3: rank 56). Thus, all baits were highly abundant in the output material arguing for efficient enrichment.

To judge the complexity of the eluates, we present the number of identified proteins in the respective eluates in the **new Supplementary Fig. 1d**. We quantified about 1,500 - 2,000 proteins in all eluates, indicating that the eluates are quite complex. It is important to note that we purposefully did not stringently purify the kinase and interacting proteins but rather base interactions on co-enrichment of interactors with the bait (also see Keilhauer, Hein and Mann, MCP, 2015).

3. On the same point, were there attempts to determine bait protein expression levels by MS and to normalize for the different HA-tagged kinase expression levels. For direct and reliable comparisons of different kinases, two HA-tagged kinases should be compared in the same experiment (e.g. kinase 1: heavy, kinase 2: light SILAC label).

The reviewer argues that it would be better to directly compare two different kinases with each other rather than comparing them individually to the empty vector control. This would indeed be an alternative way how the experiments could be performed. However, we think that the “vector control” approach we took is preferable for two reasons. First, there are 21 possible pairwise combinations of the 7 kinases, which would require $21 * 3$ replicates = 63 pulldown experiments. Second, as pointed out in our response to point 2 above, the kinase expression level does not appear to play a major role for the interaction partners detected.

In order to experimentally assess whether this approach can reveal interactors compared to our “vector control” approach, we followed the reviewers suggestion and performed the experiment. Here, we transfected SILAC L and H cells in label-swap duplicates with the kinase of HHV5 (pUL97) and HHV4

(BGLF4) (see new Supplementary Fig. 3a-b). We specifically looked at proteins that our initial analyses identified as very selective to the BGLF4 and pUL97 kinases (see Fig. 1c, clusters labeled with “chromatin silencing” ,“DNA replication” and “Sumoylation of transcription factors”, “Polycomb repressive complex 1”). We found that proteins in these clusters also showed up as being enriched with the respective kinase when directly comparing pUL97 to BGLF4. Thus, our initial analysis delivers results consistent with direct kinase to kinase comparisons.

4. Given the proposed interaction mode of the cyclin A with certain CHPKs, it is surprising that human CDK1 and CDK2 were identified as specific interactors as well. What is this explanation? Is there a ternary complex of CHPK-cyclinA-CDK1/2 or is cyclin A binding to a CHPK mutually exclusive to binding to CDK1/2?

The reviewer is right. The CHPK forms higher-order complexes with cyclins/CDKs as shown throughout our manuscript (Fig. 1c-d and 3a). We do not think that this is surprising since the CHPK interacts with cyclin A on the basis of the RXL-motif. RXL-motifs are typically present in cellular substrates of cyclin A-CDK and interact with a region in Cyclin A (“hydrophobic patch”). This region differs from the CDK interaction domain (“Cyclin box”). Thus, the CHPK does not replace the CDK in the cyclin-CDK complex but rather mimics a cyclin A-CDK substrate that can exist in a higher-order complex with cyclin-CDKs.

5. lines 167-176: The conclusions from the experiments with the predicted NLS motif transplanted into a reporter construct (Fig 2 c-d) should be toned down: NLSs can be regulated in multiple ways and the presented data is at best indicative of a functional NLS. It is by no means ‘confirming’ (line 171), in particular as the RXL to AXA mutation only leads to a partial phenotype in HHV5-U69.

We re-phrased our statement in the revised version of the manuscript as:

“Integration of wild type (WT) NLS-RXL regions into the chimeric reporter induced nuclear accumulation of the otherwise cytoplasmic GFP signal (Fig. 2d), indicative of a functional NLS.”

6. All immunoblot experiments and gels should contain molecular weight markers.

We added molecular weight markers to all immunoblots throughout the manuscript.

7. For the immunofluorescence experiments, the authors should add statements how cells were selected and if the conclusions are only based on the few cells shown in the figure panels or if quantification was done.

Following the referees comment we added quantitative data for all microscopic experiments throughout the manuscript (Fig. 2e, Fig. 4b, Fig. 5e). We also added a clarification to the methods section:

“For quantification, the microscope slides were randomly scanned and all cells were counted in these randomly acquired frames.”

REVIEWERS' COMMENTS:

Reviewer #1 (Remarks to the Author):

The authors carefully addressed all the concerns. This study significantly expands our understanding of viral protein kinases interaction with host proteins, which provides a foundation for future detailed study.

Several minor points were listed for clarification:

1. For all SILAC experiments, the sample preparation order should be detailed in method and/or figure legend. For example, are the labeled cells mixed first and then lysed together, and then IP was performed?
2. Line 507: Was false discovery rate calculated based on target-decoy database?
3. Line 544; "FDR calculations were based on the simulated data as false positives". Could the authors add a reference on how simulated data were generated?
4. Supplemental Figure 6 b-e; the "M97 IP or M97 AP" label on top of each panel should be changed to "HA-M97 AP".

Reviewed by Renfeng Li

Reviewer #2 (Remarks to the Author):

The authors effectively addressed all comments from all reviewers.

Reviewed by Rob Kalejta, University of Wisconsin-Madison

Reviewer #3 (Remarks to the Author):

The authors have answered the reviewer comments with satisfaction by conducting several additional experiments and data analysis. The work is novel and of broad interest to the Nature Communications readership.

Oliver Hantschel

Point-by-Point Response to Reviewer Comments

Reviewer #1 (Remarks to the Author):

Several minor points were listed for clarification:

1. For all SILAC experiments, the sample preparation order should be detailed in method and/or figure legend. For example, are the labeled cells mixed first and then lysed together, and then IP was performed?
2. Line 507: Was false discovery rate calculated based on target-decoy database?
3. Line 544; "FDR calculations were based on the simulated data as false positives". Could the authors add a reference on how simulated data were generated?
4. Supplemental Figure 6 b-e; the "M97 IP or M97 AP" label on top of each panel should be changed to "HA-M97 AP".

Response to Reviewer #1

1. *We added two clarifying sentences to the Methods part (lines 441-443).*
2. *This is correct. We added a clarification to the corresponding section (lines 499-500).*
3. *This permutation-based estimation of the FDR is similar to the way the Perseus software handles interaction proteomics data. We therefore cite now Tyanova et al., Nature Methods, 2016 (line 538).*
4. *Thanks for catching. We changed the labels.*